# A Mixture of Exemplars Approach for Efficient Out-of-Distribution Detection with Foundation Models

**Evelyn J. Mannix**  *evelyn.mannix@unimelb.edu.au*
*School of Mathematics and Statistics*
*University of Melbourne*

**Howard Bondell**  *howard.bondell@unimelb.edu.au*
*School of Mathematics and Statistics*
*University of Melbourne*

**Reviewed on OpenReview:** *https://openreview.net/forum?id=xpKqnSJtE4*

## Abstract

One of the early weaknesses identified in deep neural networks trained for image classification tasks was their inability to provide low confidence predictions on out-of-distribution (OOD) data that was significantly different from the in-distribution (ID) data used to train them. Representation learning, where neural networks are trained in specific ways that improve their ability to detect OOD examples, has emerged as a promising solution. However, these approaches require long training times and can add additional overhead to detect OOD examples. Recent developments in Vision Transformer (ViT) foundation models—large networks trained on large and diverse datasets with self-supervised approaches—also show strong performance in OOD detection, and could address these challenges. This paper presents Mixture of Exemplars (MoLAR), an efficient approach to tackling OOD detection challenges that is designed to maximise the benefit of training a classifier with a high quality, frozen, pretrained foundation model backbone. MoLAR provides strong OOD detection performance when only comparing the similarity of OOD examples to the exemplars, a small set of images chosen to be representative of the dataset, leading to significantly reduced overhead for OOD detection inference over other methods that provide best performance when the full ID dataset is used. Extensive experiments demonstrate the improved OOD detection performance of MoLAR in comparison to comparable approaches in both supervised and semi-supervised settings, and code is available at github.com/emannix/molar-mixture-of-exemplars.

## 1 Introduction

From managing weeds (Olsen et al., 2019), to food safety (Sandberg et al., 2023), to diagnosing cancer (Khellaf et al., 2023), deep learning is used extensively in computer vision applications where identifying out-of-distribution (OOD) data, such as plants or food contaminants not present in the training dataset, can be of great importance. While the original methods of training neural networks from scratch resulted in poor correlations between confidence and OOD data (Nguyen et al., 2015), modern foundation models (Oquab et al., 2024)—models trained on large and diverse datasets using self-supervised learning methods that offer quick training and strong performance even when the backbone is frozen—can achieve sound OOD detection results by comparing the distances between in-distribution (ID) and OOD image embeddings.

These distance-based OOD detection metrics have been well developed in the literature (Sun et al., 2022), and they can be improved by using techniques that improve the embeddings of the network for OOD detection. For example, CIDER (Ming et al., 2023) improves OOD detection performance by learning an embedding that tightly clusters classes while also maximising the distance between them, using a von Mises-Fisher

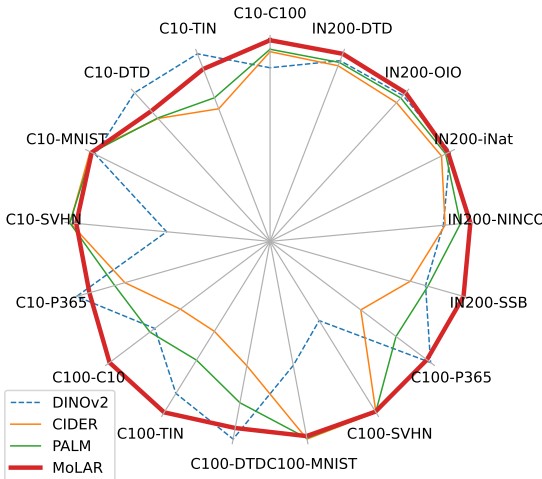

Figure 1: **Effectiveness of OOD detection with exemplars.** Comparison of Mixture of Exemplars (MoLAR) against comparable approaches, trained using a frozen DINOv2 ViT-S/14 foundation model backbone (Oquab et al., 2024). MoLAR outperforms PALM and CIDER across most in-distribution—out-of-distribution pairs, and improves on the DINOv2 backbone, particularly for datasets not within the DINOv2 training dataset (LVD-142M).

(vMF) mixture modelling approach. PALM (Lu et al., 2024) extends on this method to enable each class to be modelled by multiple mixture components, further improving OOD detection performance.

While effective, these methods find it challenging to learn the vMF cluster centres and require long training times. In CIDER (Ming et al., 2023), the cluster centres are not directly optimised using backpropogation, but are instead learned using an Exponential Moving Average (EMA) approach. PALM (Lu et al., 2024) also learns the cluster centers using this method, and assigns the mixture components to images within a batch using the iterative Sinkhorn-Knopp algorithm, adding further complexity. PALM and CIDER also do not use the learned cluster centers for OOD detection, and instead employ a KNN (Sun et al., 2022) or Mahalanobis (Lee et al., 2018) OOD metric based on a sample of the ID training data. This is a missed opportunity in making OOD detection more efficient. If distance to the vMF clusters could be used to identify OOD examples effectively, there would be no additional cost associated with using the model to both classify images and also screen for OOD data.

It is also unclear whether these approaches are able to synergise with foundation model backbones. PALM (Lu et al., 2024) and CIDER (Ming et al., 2023) were both designed to train a convolutional neural network (CNN) backbone with a projection head—a MLP network with a few layers—and their performance has not been previously tested with a frozen pretrained Vision Transformer (ViT) (Dosovitskiy et al., 2021). Nevertheless, there are other similar deep learning approaches that could also be adapted to OOD detection, that can train vMF cluster centres within a mixture model framework simply and efficiently. In the PAWS (Assran et al., 2021) semi-supervised learning approach, cluster centres are defined by the embeddings of a small set of labelled data and models are trained using a consistency based loss.

To solve these challenges, this paper presents Mixture of Exemplars (MoLAR), an OOD detection approach inspired by PAWS (Assran et al., 2021) that uses *exemplars*—particular images within the training dataset that define the centres of the vMF mixture model components—to obtain state-of-the-art results in OOD detection. MoLAR is able to obtain strong performance (Fig. 1) using only distances from exemplars to determine if a point is OOD, making testing for OOD data free and efficient in comparison to other methods. Further, MoLAR provides a unified way of tackling OOD detection challenges in both supervised and semi-supervised settings, while still achieving competitive performance. In this paper, we:

- Describe MoLAR, an approach to OOD detection that can handle *both* supervised and semi-supervised settings and is designed to be trained with a frozen foundation model backbone. Unlike previous comparable approaches such as PALM and CIDER, MoLAR is designed to identify OOD examples in a way that is *no more computationally expensive than making a class prediction.*

- Demonstrate *MoLAR obtains state-of-the-art OOD detection performance* in both supervised and semi-supervised settings employing the OpenOOD (Yang et al., 2022) benchmarks. Further, we find MoLAR obtains similar performance when only exemplars are used for OOD detection, which provides for significantly reduced overhead for OOD inference. Other methods such as PALM and CIDER have reduced performance in this case.

- Show that MoLAR-SS, the first semi-supervised method that obtains similar OOD detection performance in comparison to supervised approaches, obtains *competitive accuracy with state-of-the-art semi-supervised learning models.* Further, the design of MoLAR-SS contributes to strong semi-supervised accuracy across a broader range of datasets.

## 2 Related work

**Out-of-distribution detection.** The problem of out-of-distribution (OOD) detection in deep learning was introduced when it was observed that neural networks tend to provide overconfident results when making predictions on images with previously unseen classes or that are unrecognizable (Nguyen et al., 2015). OOD detection methods are designed to detect when inference is going to be made on such data, so an appropriate alternative action can be taken (Tao et al., 2023).

Representational approaches such as PALM (Lu et al., 2024) and CIDER (Ming et al., 2023) are designed to learn neural networks that produce an embedding of an image that more accurately identifies OOD images under particular OOD detection metrics. Key metrics include KNN (Sun et al., 2022), which uses the distance of a point to its $k^{\text{th}}$ nearest neighbour in a sample of the training dataset; and the Mahalanobis distance (Lee et al., 2018), a measure of distance between a point in high dimensional space and a distribution derived from the training data, that uses class means and a pre-calculated covariance matrix. Other works improve OOD detection in a variety of ways, from using particular types of image augmentations (Hendrycks et al., 2022), employing ensembles (Lakshminarayanan et al., 2017) to adding corrections to the logits to better capture OOD examples (Wei et al., 2022).

This paper focuses on improving supervised representational OOD detection methods, like PALM and CIDER, where annotated ID labelled data is available for training. In this context benchmarking OOD detection approaches can be challenging, as there are a large number of different ID and OOD dataset pairs that can be used. This paper follows the OpenOOD benchmarks (Yang et al., 2022), to ensure better comparability with other works in this space. While many other works in representational OOD detection focus on using self-supervised approaches (Tack et al., 2020; Li et al., 2024a; Seifi et al., 2024; Isaac-Medina et al., 2024), vMF clustering approaches like PALM and CIDER provide an intuition for *why* these representational approaches work—through tightly clustering in-distribution data. It is explored in this work if these techniques can be used in conjunction with foundation models, which are trained using self-supervised approaches and have strong OOD detection performance, for further improvements.

**Foundation models.** In computer vision, foundation models can be used as functional maps from the image domain to a representation space that captures the semantic content of the image with minimal loss of information. The degree to which semantic content is retained can be tested by undertaking downstream tasks within the representation space, such as image classification, segmentation and object detection (Rani et al., 2023). Current state-of-the-art foundation models include methods that combine several self-supervised approaches on large datasets, such as DINOv2 (Oquab et al., 2024), and other approaches that use language-image pretraining approaches such as CLIP (Radford et al., 2021; Cherti et al., 2023).

**Semi-supervised learning.** In semi-supervised learning (SSL), a model is trained to perform a particular task, such as image classification, using a small set of labelled data and a much larger set of unlabelled data. By taking into account the distribution of the unlabelled data within the learning process, SSL methods can

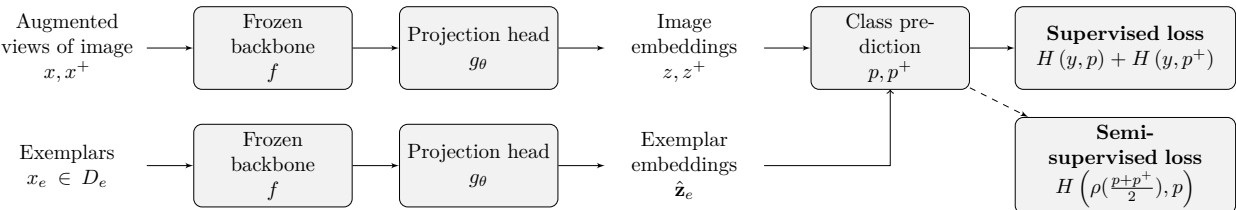

Figure 2: **Training framework for MoLAR.** Using a vMF mixture modelling approach within the image embeddings, MoLAR makes classifications on the basis of a set of exemplars with known labels. In the supervised case, a cross-entropy loss is employed. In the semi-supervised case, predictions are averaged and sharpened to provide a strong supervision signal.

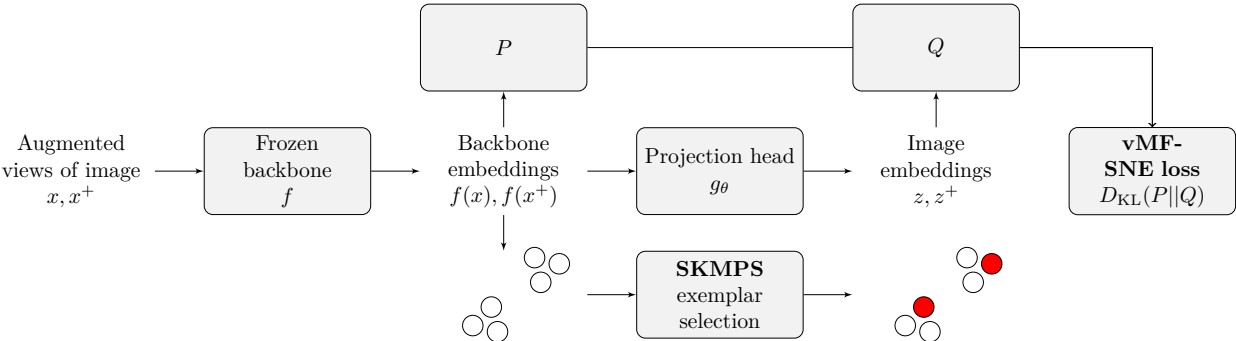

Figure 3: **Initialisation framework for MoLAR.** The SKMPS strategy is used to select representative exemplars from the dataset. The vMF-SNE approach is used to initialise the embeddings of the projection head—to reflect the local structure of the high quality foundation model embeddings.

train more accurate models compared to approaches that only utilise the labelled data within this context (Van Engelen & Hoos, 2020). The earliest SSL approaches used pseudolabelling (Lee et al., 2013), and a number of modern approaches have extended on this idea such as FixMatch (Sohn et al., 2020), which employs image augmentation to efficiently propogate labels. Further methods have improved on FixMatch by better accounting for the geometry of the image embeddings (Li et al., 2024b; Wang et al., 2022c; Cai et al., 2022; Yang et al., 2023). Of all these approaches, PAWS (Assran et al., 2021) has striking similarities to CIDER and PALM. All three methods use a vMF mixture modelling approach, though where PAWS uses the embeddings of the labelled data to generate cluster centers, CIDER and PALM introduce them as learnable weights. Notably, where CIDER and PALM are designed to learn a compact representation of the training data, PAWS uses a consistency based loss where labels are swapped between different views of an image to provide a semi-supervised learning signal.

## 3 Methodology

**Motivation.** Using exemplars—images selected from the training data whose embeddings define vMF cluster centres—offers a number of advantages over directly introducing cluster centre embeddings as additional learnable weights, as is done in PALM and CIDER. These advantages include (i) exemplars have a strong initialisation, being defined by data itself, (ii) exemplars can be augmented in the same fashion as the training data, which prevents mismatch under augmentation, (iii) exemplar embeddings can still be fine-tuned as their embeddings under the image of a neural network are used, allowing their positioning relative to other images to change and (iv) the position of exemplar embeddings is automatically changed with each gradient update to the network, allowing them to be efficiently learnt through backpropagation. This latter point removes the need for cluster centre embeddings to be learned using the EMA approach employed by PALM and CIDER.

**Notation.** This paper considers an OOD detection setting, with an ID dataset $D$ which is used to train the model, with the aim of being able to detect unseen OOD data $D^*$. In this work, the ID data has a subset of points called exemplars $D_e \subset D$, which can be selected randomly or using a particular strategy. We further consider a frozen pretrained transformer $f$ and a trainable projection head $g_\theta$, where the parameters $\theta$ are optimised using stochastic gradient descent on batches of images with labels $\{x, y\} \in D$. For an image batch we use $x_i, y_i$ to refer to specific examples.

**Supervised MoLAR.** We define MoLAR as a mixture model with von Mises-Fisher components

$$p(\hat{z}|D_e) := \frac{1}{|D_e|} \sum_{x_i \in D_e} k(\hat{z}; \hat{z}_i) \tag{1}$$

$$k(\hat{z}_j; \hat{z}_i) \propto \exp\left(\hat{z}_j \cdot \hat{z}_i / \tau\right) \tag{2}$$

where $z_i = g_\theta(f(x_i))$. Inputs are normalized onto the unit sphere $\hat{z}_j = \frac{\hat{z}_j}{||\hat{z}_j||}$ and $\tau$ is a temperature hyperparameter that sets the diffuseness of the distribution. For a particular class $y$, we have

$$p(\hat{z}|y, D_e) = \frac{1}{|D_e^y|} \sum_{x_i \in D_e^y} k(\hat{z}; \hat{z}_i) \tag{3}$$

where $D_e^y$ are the exemplars of class $y$. Using Bayes' rule provides a way to classify new examples $z$ with

$$p(y|\hat{z}, D_e) = \frac{p(\hat{z}|y, D_e)p(y)}{p(\hat{z})} \tag{4}$$

$$= \sigma(\hat{z} \cdot \hat{\mathbf{z}}_e / \tau) \cdot \boldsymbol{\phi} \tag{5}$$

where $\sigma(\cdot)$ is the softmax function, we assume a uniform prior $p(y)$, and use $\hat{\mathbf{z}}_e$ to refer to the matrix of the normalized exemplar embeddings. Here $\boldsymbol{\phi}$ is the matrix of the one-hot encoded exemplar class labels. This provides for a straightforward maximum log-likelihood estimation approach for training the model

$$\mathcal{L}_{\text{supervised}}(D, D_e; \theta) := -\frac{1}{|D|} \sum_{x_i, y_i \in D} \log p(y_i|\hat{z}_i, D_e) \tag{6}$$

which is equivalent to a cross entropy loss.

***Related methods: PALM and CIDER.*** MoLAR uses the same model definition as CIDER (Sun et al., 2022) and PALM (Lu et al., 2024), but rather than the vMF cluster centres being introduced as learnable weights, they are defined by the embeddings of the exemplars in $D_e$. Further, MoLAR does not include the second loss term in PALM and CIDER that encourages separation of the cluster centres for different classes. This term was found not to be necessary to obtain strong OOD performance with MoLAR as Eq. (6) already encourages classes to be well separated, and it was also found to hurt classification accuracy (Table S16).

**Semi-Supervised MoLAR.** In the semi-supervised setting, there are only labels $y$ for the exemplars $D_e$. We introduce semi-supervised MoLAR (MoLAR-SS) to deal with this case. Through adapting the PAWS (Assran et al., 2021) semi-supervised learning approach, it is found that averaging and sharpening predictions over multiple augmented views $x, x^+$, similar to MixMatch (Berthelot et al., 2019), provides strong performance

$$\mathcal{L}_{\text{semi-supervised}} := \frac{1}{2|D|} \sum_{i=1}^{|D|} \left( H\left(\rho\left(\frac{p_i + p_i^+}{2}\right), p_i\right) \right. \tag{7}$$

$$\left. + H\left(\rho\left(\frac{p_i + p_i^+}{2}\right), p_i^+\right) \right) - H(\overline{p}),$$

where $p_i = p(y|\hat{z}_i, D_e)$ as given by Eq. (5), and $H(p, q) = -\sum_i p_i \log q_i$ is cross-entropy loss. The function $\rho(\cdot)$ is a sharpening function with temperature hyperparameter $T$

$$\rho(p)_i := \frac{p_i^{1/T}}{\sum_{i=1}^{C} p_i^{1/T}} \tag{8}$$

and in the final expression, $\bar{p} = \sum_i^{|D|} \rho\left(\frac{p_i + p_i^+}{2}\right)$ is the average of the sharpened predictions on the unlabelled images, which encourages the predictions to be balanced across the classes.

***Related methods: PAWS.*** MoLAR-SS is similar to PAWS (Assran et al., 2021; Mo et al., 2023) with several key differences. These include (i) replacing the original consistency based loss with an averaged and sharpened loss; (ii) removing the additional classification head, reporting accuracy only using Eq. (5); and (iii) considering exemplar selection strategies and initialising the projection head with the foundation model embeddings, as outlined in the next two sections. These contributions provide improved semi-supervised accuracy and OOD detection performance in the context of training with a frozen foundation model backbone.

**Exemplar selection strategies.** In this step, the selection of exemplars $D_e$ from the whole dataset $D$ is considered. The goal is to select the set of exemplars that are most representative of the data as evidenced by better downstream accuracy and OOD detection performance. In the default case, exemplars are generally chosen randomly, stratifying by class so that an equal number of images from each class are selected. However, this requires all the data to be labelled in the first instance, which is at odds with a semi-supervised learning context.

This work introduces a simple $k$-means exemplar selection (SKMPS) strategy. This approach provides for a diverse sampling of exemplars from a given training dataset in three steps: (i) calculate the normalised foundation model embeddings of the images, then (ii) cluster them into $k$ clusters, where $k$ is the desired number of exemplars (i.e. number of labelled images in a semi-supervised setting). And (iii) select the image closest to the centroid of each cluster as an exemplar.

We choose $k$-means clustering (Lloyd, 1982) for the clustering algorithm in the second step as it is scalable and allows for the selection of a fixed number of clusters. The $k$-means algorithm minimises the sum of squared euclidean distances to the cluster centroids (Lloyd, 1982). By normalizing the image embeddings given by the foundation model in the first step, the squared euclidean distance becomes proportional to cosine distance, which is used for determining image similarity with foundation models (Oquab et al., 2024; Radford et al., 2021). In this sense, the SKMPS procedure can be interpreted as identifying clusters of visually similar images, and selecting the most representative image from each cluster as an exemplar.

***Related methods: USL.*** The SKMPS exemplar selection strategy is similar to Unsupervised Selective Labelling (USL) (Wang et al., 2022b), which is proposed to improve semi-supervised learning performance by selecting a better set of labelled images in the first instance. The key difference between the two approaches is that SKMPS does not require estimating the density of the image embeddings from the dataset, which USL uses to select images from the denser regions of each cluster. This makes SKMPS simpler to apply and more computationally efficient, while simultaneously sampling a greater diversity of images in comparison to USL (Fig. 6).

**Projection head initialisation.** If a pretrained backbone model $f$ is used, which provides a high quality representation of the dataset, a *good* starting point for the projection head $g_\theta$ should reflect the local structure of the data under the image of $f$. This is not a given if the weights of the projection head are randomly initialized, and without placing restrictions on its architecture and the dimensionality of the latent space of $f$ and $g_\theta$, the projection head cannot be initialised as an identity mapping. For example, the projection head architecture employed in PAWS, and also this work, is a three layer Multi-Layer Perceptron (MLP) network with ReLU activations and batch-normalisation. This introduces non-linearity into the network which will alter the input embeddings even if the weights of each layer are initialised using the identity.

Instead, a Stochastic Neighbour Embedding (SNE) (Hinton & Roweis, 2002) approach is proposed. This involves building two probability distributions $P, Q$ using a kernel that describes the similarity of data points in the origin space (under the image of $f$) and the target space (under the image of $g_\theta \circ f$). The idea is for images $x_i, x_j$ in $D$, $p_{ij} \in P$ will have high probability if they are similar and low probability if they are dissimilar. The target distribution $Q$ in the target space can then be fit to the distribution in the origin

space $P$ by minimizing the KL divergence

$$D_{\mathrm{KL}}(P||Q) = \sum_{ij} p_{ij} \log \frac{p_{ij}}{q_{ij}} \tag{9}$$

where $p_{ij}$ and $q_{ij}$ describe the similarity between points in the origin and target space. Inspired by the definition of MoLAR as a vMF mixture model, a vMF kernel is used to calculate $P, Q$ as described in Appendix A, providing an effective method of initialising the weights of the projection head to reflect the foundation model embeddings.

## 4 Experiments

### 4.1 Experimental setup

**OOD benchmarks.** The experiments closely follow the OpenOOD (Yang et al., 2022) benchmarks, which use a specific set of near-OOD datasets and far-OOD datasets for each ID dataset. For CIFAR-10, the near-OOD datasets are CIFAR-100 (Krizhevsky et al., 2009) and Tiny ImageNet (Le & Yang, 2015). The far-OOD images are Places365 (Zhou et al., 2017a), DTD (Cimpoi et al., 2014), MNIST (Deng, 2012) and SVHN (Netzer et al., 2011). These are the same for CIFAR-100, which instead uses CIFAR-10 as a near-OOD dataset. For ImageNet-200, the near-OOD datasets are SSB-hard (Bitterwolf et al., 2023) and NINCO (Vaze et al., 2022), and the far-OOD images come from iNaturalist (Van Horn et al., 2018), OpenImage-O (Wang et al., 2022a) and DTD (Cimpoi et al., 2014).

**OOD detection metric.** A KNN OOD metric (Sun et al., 2022) is used throughout the paper, which measures OOD distance for an example image based on the closest training examples in the normalized embedding space. The KNN OOD metric for an example image $x$ is given by cosine distance

$$G(x; \alpha, k) = 1 - \left\| \hat{z} \cdot \widehat{\mathrm{NN}}_{k, g \circ f}(x; D_\alpha) \right\| \tag{10}$$

where $\widehat{\mathrm{NN}}_{k, g \circ f}$ is the normalised embeddings of the $k^{\mathrm{th}}$ nearest neighbour to $x$ in $D_\alpha$ within the feature space of the network $(g \circ f)$. The parameter $\alpha$ is the proportion of the training dataset that is used for finding the nearest training examples $(D_\alpha \subseteq D)$. Best performance for the KNN metric is generally reported using $\alpha = 1.0$ (the whole training dataset is used) and $k = 1$ (only the distance to the closest in-distribution training point is used) (Sun et al., 2022). These are the KNN hyperparameters that are used throughout this paper, unless stated otherwise. Computing the KNN metric is much more efficient with a smaller $\alpha$, at the cost of performance, and we explore using the exemplar points $D_e$ selected with SKMPS instead of $D_\alpha$ to reduce OOD detection overhead with minimal performance loss.

The performance of OOD detection is measured using the Area Under the Receiver Operating Characteristic curve (AUROC), employing this KNN OOD distance metric to classify between ID and OOD examples, as done in previous works (Yang et al., 2022).

**Semi-supervised benchmarks.** Several semi-supervised learning benchmarks are also considered to test the performance of MoLAR-SS. We focus on the well-studied CIFAR-10 (10 classes), CIFAR-100 (100 classes) (Krizhevsky et al., 2009) and Food-101 (101 classes) (Bossard et al., 2014) benchmarks, but also include some contextual results on EuroSAT (10 classes) (Helber et al., 2019), Flowers-102 (102 classes) (Nilsback & Zisserman, 2008), Oxford Pets (37 classes) (Parkhi et al., 2012) and DeepWeeds (2 classes) (Olsen et al., 2019). While DeepWeeds has originally 9 classes, we treat it as a binary classification problem to identify weeds versus not weeds, due to the size of the negative class. For EuroSAT and DeepWeeds, where preset validation splits were not available, we sampled 500 images per class, and 30% of the images for the 9 original classes respectively.

**Exemplar selection.** When sampling exemplars using SKMPS, USL or using a random class-stratified sample, 4 (CIFAR-10), 4 (CIFAR-100), 4 (EuroSAT), 27 (DeepWeeds - binary), 2 (Flowers-102), 2 (Food-101), 2 (Oxford Pets) and 3 (ImageNet-200) images per class are selected. The size of these sets of exemplars

are selected to be in line with previous semi-supervised learning results, and also sufficient to sample every class as explored for key datasets in Fig. S9 in the supporting information. Due to the noise in the results when reporting accuracy in semi-supervised learning settings, we report the mean of five runs and the standard deviation for all datasets except ImageNet-200.

**Hyperparameter selection.** For CIDER, PALM and PAWS, it was found that the model specific hyperparameters used with ResNet backbones also worked well when using a DINOv2 backbone. For MoLAR the same hyperparameters as PAWS were used. Other hyperparameters, such as batch size, learning rates, schedulers, optimisers and image augmentations, were kept the same between all models and are detailed in full in the github repository. An ablation study was further undertaken on the hyperparameters introduced as part of the vMF-SNE initialisation step in Appendix C.2 in the supporting information. The hyperparameter configurations for CIDER and PALM followed the differences these methods used across datasets, whereas the hyperameters for MoLAR were unchanged across datasets.

## 4.2 Results

Table 1: **OOD detection.** Comparison of representational approaches using a DINOv2 ViT-S/14 frozen backbone on OpenOOD benchmarks (Yang et al., 2022) with a KNN metric (Sun et al., 2022). We use — to refer to the performance of the backbone embeddings (without a projection head).

| Backbone | Methods | Datasets | | | | | |
|---|---|---|---|---|---|---|---|
| | | CIFAR-10 | | CIFAR-100 | | IN-200 | |
| | | Near-OOD | Far-OOD | Near-OOD | Far-OOD | Near-OOD | Far-OOD |
| ResNets (finetuned) | CIDER (Ming et al., 2023; Lu et al., 2024) | 90.7 | 94.7 | 72.1 | 80.5 | 80.58 | 90.66 |
| | PALM (Lu et al., 2024) | 93.0 | 98.1 | 76.9 | 93.0 | | |
| DINOv2 ViT-S/14 (frozen) | — | **95.8** | 98.1 | 89.9 | 92.2 | 86.3 | 93.1 |
| | CIDER | 92.4 | 97.6 | 76.5 | 91.2 | 86.7 | 96.2 |
| | PALM | 94.3 | 98.6 | 86.4 | 95.7 | 89.1 | **96.7** |
| | MoLAR | 95.4 | **99.1** | **94.9** | **97.4** | **90.7** | **96.7** |

**MoLAR is competitive with previous OOD detection methods.** Table 1 shows that MoLAR provides strong performance in comparison to previous methods, particularly for near-OOD datasets. Overall, CIDER and PALM have improved performance with a frozen DINOv2 backbone, but they can struggle to match the performance of the backbone itself on near-OOD detection for some datasets. Conversely, MoLAR is able to significantly improve on the near-OOD performance of the backbone in every case—with the exception of near-OOD CIFAR-10—providing particularly strong performance on CIFAR-100.

Table 2: **OOD detection with exemplars.** Comparison of representational approaches using a DINOv2 ViT-S/14 frozen backbone on OpenOOD benchmarks (Yang et al., 2022) with a KNN metric (Sun et al., 2022), employing a consistent set of exemplars obtained with SKMPS. We select 40 exemplars for CIFAR-10, 400 for CIFAR-100 and 600 for IN-200. We use — to refer to the performance of the backbone embeddings (without a projection head), and bold the best fully supervised and the best semi-supervised methods separately to reflect that semi-supervised OOD detection is a more challenging problem. Cases are left unbolded if they do not improve upon the backbone embeddings.

| Backbone | Methods | | Datasets | | | | | |
|---|---|---|---|---|---|---|---|---|
| | | | CIFAR-10 | | CIFAR-100 | | IN-200 | |
| | | | Near-OOD | Far-OOD | Near-OOD | Far-OOD | Near-OOD | Far-OOD |
| DINOv2 ViT-S/14 (frozen) | Self-supervised | — | 93.9 | 94.7 | 88.3 | 89.2 | 87.1 | 96.4 |
| | Supervised | CIDER | 90.7 | 96.8 | 78.6 | 91.1 | 86.1 | 95.5 |
| | | PALM | 91.7 | 97.1 | 85.2 | 94.5 | 88.2 | 96.0 |
| | | MoLAR | **94.4** | **98.0** | **95.5** | **97.1** | **91.4** | **96.7** |
| | Semi-supervised | PAWS | 82.6 | 90.5 | 90.6 | 94.5 | 88.9 | 95.1 |
| | | MoLAR-SS | 93.1 | **98.0** | **95.2** | **96.3** | **92.5** | **96.3** |

Table 3: **Timings and OOD detection performance comparisons.** Combined results for MoLAR and PALM from Table 1 and Table 2 with timing comparisons between using all training data for the KNN metric, and the SKMPS subset.

| | | | | Method | | | |
| | | | | MoLAR | | PALM | |
| Dataset | KNN Set | Images | OOD Overhead (ms) | Near-OOD | Far-OOD | Near-OOD | Far-OOD |
|---|---|---|---|---|---|---|---|
| CIFAR-10 | All | 50,000 | 0.15 | 95.4 | 99.1 | 94.3 | 98.6 |
| | SKMPS | 40 | 0.0002 | 94.4 | 98.0 | 91.7 | 97.1 |
| CIFAR-100 | All | 50,000 | 0.15 | 94.9 | 97.4 | 86.4 | 95.7 |
| | SKMPS | 400 | 0.001 | 95.5 | 97.1 | 85.2 | 94.5 |
| ImageNet-200 | All | 258,951 | 0.75 | 90.7 | 96.7 | 89.1 | 96.7 |
| | SKMPS | 600 | 0.002 | 91.4 | 96.7 | 88.2 | 96.0 |

**MoLAR retains strong performance when only using exemplars to detect OOD examples.** Table 2 shows the performance of different OOD detection approaches when the set of exemplars selected by SKMPS is used as the ID data. MoLAR retains strong OOD detection performance, and for near-OOD CIFAR-100 and ImageNet-200 performance is similar to using the full training dataset as ID. The performance of CIDER and PALM fall in every case except for near-OOD CIFAR-100. We also observe strong performance for MoLAR-SS, which performs similarly to MoLAR, providing stronger performance than the fully supervised CIDER and PALM methods. Compared to using the full training dataset as ID, using exemplars leads to significantly faster OOD inference across all datasets and allows OOD detection with minimal overhead (Table 3). For MoLAR and MoLAR-SS, the distance to these exemplars is already computed in the process of making a class prediction, making OOD inference *free* in this case while still providing strong performance.

Table 4: **Semi-supervised learning.** Performance of MoLAR versus PAWS and other methods. Comparable model sizes in each column are underlined, and the best performing model of these is **bolded**. Entries noted with * use a larger labelled dataset (1% of the data) than in our benchmarks.

| Methods | Backbone | | Datasets | | |
| | | | C10 | C100 | Food |
|---|---|---|---|---|---|
| FreeMatch (Wang et al., 2022c) | WRN-28-2/8 | 1.5M/23M | 95.1 | 62.0 | |
| +SemiReward (Li et al., 2024b) | ViT-S-P4-32 | 21M | | 84.4 | |
| Semi-ViT (Cai et al., 2022) | ViT-Base | 86M | | | 82.1* |
| PAWS | DINOv2 ViT-S/14 (f) | 21M | 92.9±2.1 | 70.9±1.5 | 75.1±2.0 |
| MoLAR-SS | DINOv2 ViT-S/14 (f) | 21M | **95.9**±0.0 | 77.3±0.4 | 81.7±0.7 |
| | DINOv2 ViT-B/14 (f) | 86M | 98.1±0.1 | 85.5±0.3 | **87.2**±0.8 |
| | DINOv2 ViT-L/14 (f) | 300M | 99.2±0.0 | 89.8±0.2 | 90.1±0.8 |

**MoLAR-SS is competitive with previous semi-supervised learning methods.** Table 4 shows that across a range of benchmarking datasets MoLAR-SS outperforms PAWS using the same frozen backbone model, and is competitive with other methods in the literature that train models of similar sizes and using the same number (or greater) of labelled images. MoLAR-SS outperforms FreeMatch (Wang et al., 2022c) on CIFAR-10 by 0.8% on average with the smallest DINOv2 model available, and also outperforms Semi-ViT (Li et al., 2024b) by 5.1% using a much smaller labelled set. SemiReward (Li et al., 2024b) obtains better results with a similar model size for CIFAR-100, but in contrast to MoLAR this approach trains all model weights whereas MoLAR-SS uses a frozen backbone. MoLAR-SS outperforms SemiReward (+1.1%) with the ViT-B model, which takes about two hours of GPU time using two Nvidia V100 GPUs, compared to the 67 hours of compute time for SemiReward on a single more powerful Nvidia A100 GPU (Li et al., 2024b).

**SKMPS obtains competitive performance in comparison to USL on downstream semi-supervised classification tasks.** Table 5 shows that using exemplar selection strategies improves the performance of both PAWS and MoLAR-SS, and that the best performing approach across both methods and all three datasets considered is the SKMPS strategy. While the results between USL (Wang et al., 2022b) and SKMPS are similar for CIFAR-100, significantly better results are obtained on the CIFAR-10

Table 5: **Exemplar selection strategies.** Comparison of different exemplar selection strategies with MoLAR-SS and PAWS.

| Method | Exemplar sel. strat. | Datasets | | |
|---|---|---|---|---|
| | | C10 | C100 | Food |
| PAWS | Random Strat. | 92.9±2.1 | 70.9±1.5 | 75.1±2.0 |
| | USL | 93.6±0.4 | **74.4**±0.6 | 79.1±1.1 |
| | SKMPS | **94.4**±0.5 | **74.8**±0.3 | **80.5**±0.6 |
| MoLAR-SS | Random Strat. | **95.8**±0.1 | 75.7±0.9 | 75.9±1.8 |
| | USL | **95.8**±0.1 | **77.5**±0.4 | 80.0±1.0 |
| | SKMPS | **95.9**±0.0 | **77.3**±0.4 | **81.7**±0.7 |

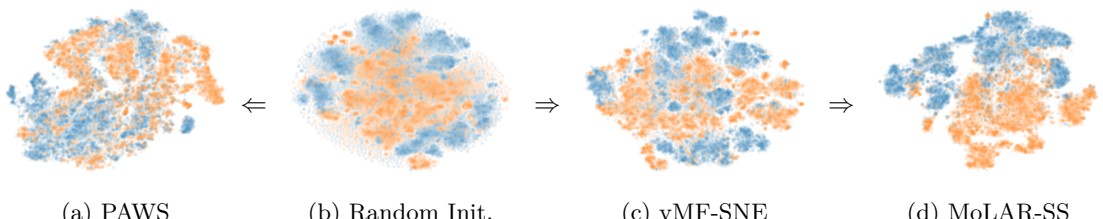

(a) PAWS  ⇐  (b) Random Init.  ⇒  (c) vMF-SNE  ⇒  (d) MoLAR-SS

Figure 4: **Visualising the impact of vMF-SNE initialisation.** *t*-SNE visualisation of the projection head embeddings at various stages of the training process for DeepWeeds. Blue is the weed present class, and orange is the negative class. The arrows represent the order of training steps.

dataset using PAWS, and on the Food-101 dataset using both PAWS and MoLAR-SS. SKMPS also provides more consistent results than USL for the Food-101 dataset, with a smaller variance in performance.

Table 6: **Semi-supervised learning on additional datasets.** Comparison of PAWS and MoLAR-SS on a wider range of benchmarks for semi-supervised learning. SKMPS is used for IN-200, while the other datasets employ a random class stratified exemplar selection strategy.

| Method | Datasets | | | | |
|---|---|---|---|---|---|
| | EuroSAT | DeepWeeds | Flowers | Pets | IN-200 |
| PAWS | 94.4±0.3 | 72.6±4.5 | 98.5±0.7 | 90.2±1.0 | 89.4 |
| MoLAR-SS | 94.2±0.1 | **87.8**±1.5 | 98.9±0.6 | **91.6**±0.2 | **91.9** |

**vMF-SNE initialisation can be important for the success of semi-supervised learning with a projection head.** In Table 6 the performance of PAWS and MoLAR-SS is explored across a broader range of datasets. In some cases, similar performance between the two approaches is found, while for others there is a drastic difference. This is particularly notable for the DeepWeeds dataset, where the vMF-SNE initialisation is important for obtaining good results. This can be seen in Fig. 4, which shows that in DeepWeeds the randomly initialised projection head clusters the data poorly in the first instance. As a result, training with PAWS results in mediocre performance and a mixing of the two classes (Fig. 4a), while training using the vMF-SNE initialised head (Fig. 4c) results in better separation (Fig. 4d) and better performance.

**SKMPS and vMF-SNE intialisation contribute to OOD detection performance.** Table 7 shows that vMF-SNE initialisation and SKMPS contribute to improving OOD performance with both MoLAR and MoLAR-SS.

**MoLAR learns compact representations.** Fig. 5 shows that MoLAR-SS learns a much more compact representation around the labelled exemplars in comparison to PAWS, which is further supported by the results in Table 2 and Table 7. This can be attributed to MoLAR-SS using an averaging and sharpening approach across views, rather than a consistency based loss, that encourages representations to be more compact while also providing superior semi-supervised learning performance. It is found that the consistency

Table 7: **Out of distribution detection ablation.** OpenOOD results for MoLAR with and without vMF-SNE initialisation and SKMPS exemplar selection. By default results are reported with these components.

| Methods | Datasets | | | |
| --- | --- | --- | --- | --- |
| | CIFAR-10 | | CIFAR-100 | |
| | Near-OOD | Far-OOD | Near-OOD | Far-OOD |
| MoLAR | 89.3 | 93.8 | 92.1 | 96.1 |
| +vMF-SNE | 89.6 | 95.0 | 94.3 | 96.7 |
| +SKMPS | **94.4** | **98.0** | **95.5** | **97.1** |
| MoLAR-SS | 91.4 | 95.6 | 93.0 | 95.4 |
| +vMF-SNE | **93.2** | 96.6 | 93.6 | 95.6 |
| +SKMPS | 93.1 | **98.0** | **95.2** | **96.3** |

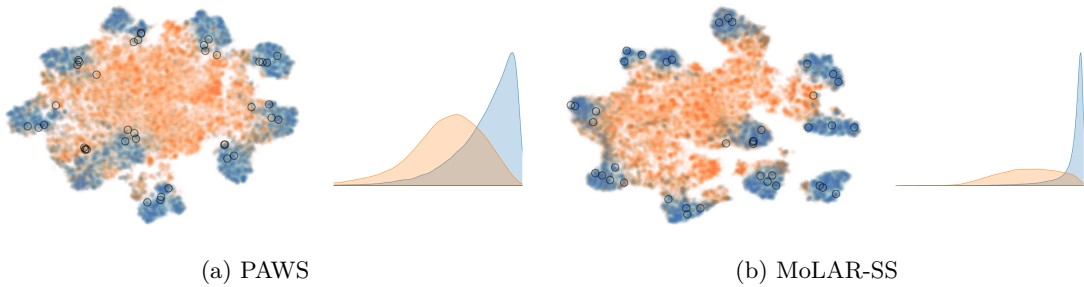

(a) PAWS  (b) MoLAR-SS

Figure 5: **Visualizing improvements in OOD detection.** $t$-SNE visualisation of the projection head embeddings and density plots of minimum cosine distance to an exemplar ($\circ$) for in-distribution CIFAR-10 (blue) and OOD CIFAR-100 (orange).

based loss can in fact result in performance degradation over a training run (Fig. S8). This is a result of different views on the class boundaries having highly confident, but different, sharpened class probabilities. For PAWS this would cause different views of the same image to be given different labels during training, resulting in instabilities at the class boundary. However, MoLAR-SS uses the same potential label for each view of an image providing a more consistent supervision signal.

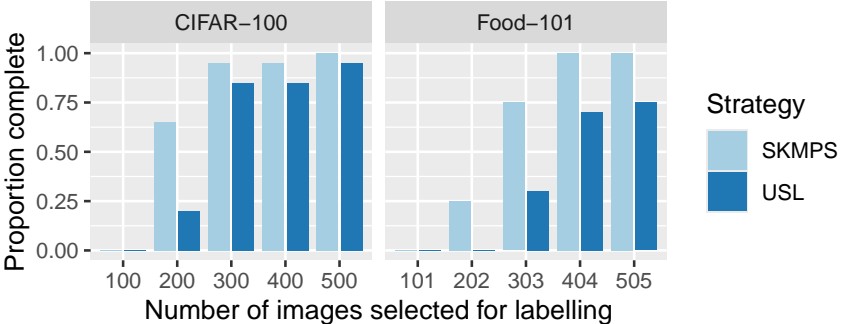

Figure 6: **Exemplar selection diversity.** Proportion of exemplar selection runs that sample all classes on different budgets $k$.

**SKMPS selects more diversity in comparison to USL.** Fig. 6 shows that SKMPS samples more classes than USL on small budgets. For example, for selecting 202 exemplars on the Food-101 dataset, USL never samples all of the classes in 20 runs whereas SKMPS samples all the classes in 25% of cases. This shows that SKMPS is better at sampling a diverse set of imagery, providing better results on downstream classification and OOD detection performance. Further results are shown Appendix B in the supporting information.

## 5 Discussion

**Relationship to prototypical approaches.** MoLAR, CIDER (Ming et al., 2023), PALM (Lu et al., 2024) and PAWS (Assran et al., 2021) can be considered prototypical deep learning approaches, which learn a metric or semi-metric space for classifying items based on their proximity to class prototypes. In this context, the exemplars used in MoLAR are equivalent to prototypes. Prototypical approaches have been used in various contexts such as improving OOD generalisation (Bai et al., 2024), robust classification (Yang et al., 2018), few-shot learning (Snell et al., 2017) and in multi-modal applications (Radford et al., 2021). They have also been applied to OOD detection in other works, to smooth logits for OOD detection (Sun et al., 2024) or to train classifiers employing a set of prototypes derived from concept similarities to improve OOD detection performance (Gong et al., 2025). This latter work is similar to the vMF clustering approach employed in CIDER, PALM and MoLAR, but rather than learning prototype representations during training, they are learnt prior using conceptual similarity in WordNet and fixed while training the neural network. Where CIDER and PALM learn prototypes non-parametrically, MoLAR refines this process by selecting images from the training dataset as exemplars using a strategy such as SKMPS, and directly learning the embeddings of these exemplars through back-propagation, as detailed in Section 3.

**Limitations.** MoLAR is designed to provide optimal performance with a foundation model backbone. While these models generalise well to many contexts, they can still offer poor performance on images that are very different from their training dataset (Zhang et al., 2024). As the performance of MoLAR for OOD detection improves on the performance of the chosen backbone, it is likely to also perform poorly in these settings. Furthermore, as MoLAR uses a frozen foundation model it is expected that for problems where a linear head fails to obtain good performance, MoLAR will also fail to perform (Chen et al., 2021).

Exemplar selection strategies such as SKMPS can be challenging to apply to training datasets with large class imbalances, or when there are a large number of classes. They can fail to adequately sample classes with a small number of training examples, and larger exemplar sets may need to be sampled in order to select atleast one exemplar from each class when more classes are present (see Fig. S9 in the supporting information). In some cases it may be better to apply SKMPS to sample a fixed number of exemplars from each class directly, if full class information is available for the given dataset.

## 6 Conclusion

This paper proposes MoLAR, a unified approach to training supervised and semi-supervised image classifiers that provides state-of-the-art OOD detection results. MoLAR is designed to be used with frozen pretrained foundation models, employing parametric vMF-SNE initialisation and simple $k$-means exemplar selection (SKMPS) to obtain the maximal benefit from the high-quality embeddings that these models make available. This enables significantly shorter training times in comparison to other OOD detection methods that finetune a CNN backbone, and MoLAR can obtain strong OOD detection performance by comparing OOD examples only to a small set of exemplars. This allows for efficient OOD detection, that is no more expensive than making a class prediction. Further, MoLAR-SS obtains competitive semi-supervised learning results in comparison to other approaches designed only to maximize performance.

## Acknowledgements

We acknowledge the Traditional Custodians of the unceded land on which the research detailed in this paper was undertaken, the Wurundjeri Woi Wurrung and Bunurong peoples of the Kulin nation, and pay our respects to their Elders past and present. This research was undertaken using the LIEF HPC-GPGPU Facility hosted at the University of Melbourne. This Facility was established with the assistance of LIEF Grant LE170100200. This research was also undertaken with the assistance of resources and services from the National Computational Infrastructure (NCI), which is supported by the Australian Government. Evelyn J. Mannix was supported by a Australian Government Research Training Program Scholarship to complete this work.

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

# Supplementary Material

## A  Parametric vMF-SNE Initialisation

This section describes the parametric Stochastic Neighbour Embedding (SNE) approach (Hinton & Roweis, 2002) used to initialize the projection head of the network in MoLAR. This method involves building two probability distributions that describe the similarity of data points in two different spaces, the origin space $P$ and the target space $Q$. The idea is for similar points $i, j$ in $P$ to have a high probability $p_{ij}$ compared to dissimilar points, and to fit the distribution in the target space $Q$ to the distribution in the origin space $P$ by minimizing the KL divergence

$$\mathrm{KL}(Q||P) = \sum_{ij} p_{ij} \log \frac{p_{ij}}{q_{ij}} \tag{11}$$

where $p_{ij}$ and $q_{ij}$ describe the similarity between points in the origin and target space.

These $P, Q$ distributions are built using a probability distribution as a kernel. As MoLAR is defined as a von Mises-Fisher (vMF) mixture model, this is a natural choice. Using the vMF kernel to compute the probabilities $P$ in the latent space defined by our backbone model $f$ provides

$$p_{j|i} := \frac{k_{\kappa_i}(\hat{h}_j; \hat{h}_i)}{\sum_{k \neq i} k_{\kappa_i}(\hat{h}_k; \hat{h}_i)}, \tag{12}$$

$$k_{\kappa_i}(\hat{h}_j; \hat{h}_i) \propto \exp\left(\hat{h}_j \cdot \hat{h}_i \, \kappa_i\right), \tag{13}$$

where $h_j = f(x_j)$ and we define $p_{i|i} = 0$, noting that the vMF normalization constant within $k_{\kappa_i}$ cancels out in this expression. We set $\kappa_i$ for each point to make the entropy $H_i$ equal to the perplexity parameter $\gamma$, $H_i = \sum_k p_{k|i} \log p_{k|i} = \log \gamma$. This ensures each point has a similar number of neighbours in the distribution $P$, and for each point $\kappa_i$ is determined using the bisection method as is standard in SNE approaches. To obtain a symmetric distribution, we use $p_{ij} = \frac{p_{j|i} + p_{i|j}}{2N}$, where $N$ is the number of data points.

To compute the probability distribution $Q$ in the latent space of the projection head, a vMF distribution with fixed concentration $\tau$, as in the main text, is used as the kernel,

$$q_{j|i} = \frac{k(\hat{z}_j; \hat{z}_i)}{\sum_{k \neq i} k(\hat{z}_k; \hat{z}_i)}, \tag{14}$$

where $k$ is a MoLAR mixture component (Eq. (2)) and we define $q_{i|j} = 0$. We also symmetrise this conditional distribution as above to obtain $q_{ij}$.

To minimize the KL divergence in Eq. (11), gradients are taken with respect to the parameters of the projection head $g_\theta$. This allows the vMF-SNE initialisation to follow a similar approach as to training the neural networks in MoLAR, utilising mini-batching, stochastic gradient descent and image augmentations (Assran et al., 2021).

This parametric vMF-SNE initialisation method is a similar to $t$-SNE (Van der Maaten & Hinton, 2008), parametric $t$-SNE (Van Der Maaten, 2009) and non-parametric vMF-SNE (Wang & Wang, 2016), but we apply it in a very different context — as an initialisation approach for neural networks using frozen foundation model embeddings. There are three key hyper-parameters in this process — the batch size, which defines the number of images to sample in each mini-batch, the perplexity parameter $\gamma$, and the concentration of the vMF distributions in $Z_2$ which is determined by $\tau$. An ablation study considering the impact of these parameters is presented Appendix C.2.

---

**Algorithm 1** vMF-SNE initialisation

---

**Input:** Training dataset $D$, $N_E$ number of epochs, $f$ backbone model, Aug(.) augmentation strategy, $\gamma$ perplexity parameter, $\tau$ temperature parameter

Randomly initialise MLP head $g_\theta$ and set starting value of $\kappa_i = 1$ for all images $i$ ;

$l = 0$;

**while** $l < N_E$ **do**

    Randomly split $D$ into $B$ mini-batches;

    **for** $(\mathbf{x}_b, \mathbf{y}_b) \in \{D_1, ..., D_b, ..., D_B\}$ **do**

        Augment $\mathbf{X} = \text{Aug}(\mathbf{x}_b)$;

        Calculate $\mathbf{H} = f(\mathbf{X})$, normalize to obtain $\hat{\mathbf{H}}$;

        Calculate $\mathbf{Z} = g_\theta(\mathbf{H})$, normalize to obtain $\hat{\mathbf{Z}}$;

        Calculate $k_{\kappa_i}(\hat{h}_j; \hat{h}_i) = \exp\left(\hat{\mathbf{H}}_{j:} \cdot \hat{\mathbf{H}}_{i:} \kappa_i\right)$ for all images $i, j$;

        Calculate $k(\hat{z}_j; \hat{z}_i) = \exp\left(\hat{\mathbf{Z}}_{j:} \cdot \hat{\mathbf{Z}}_{i:}/\tau\right)$ for all images $i, j$;

        Calculate $p_{j|i} = \frac{k_{\kappa_i}(\hat{h}_j; \hat{h}_i)}{\sum_{k \neq i} k_{\kappa_i}(\hat{h}_k; \hat{h}_i)}$;

        Calculate $q_{j|i} = \frac{k(\hat{z}_j; \hat{z}_i)}{\sum_{k \neq i} k(\hat{z}_k; \hat{z}_i)}$;

        Find optimal values for $\kappa_i$ by minimising $\left|\sum_k p_{k|i} \log p_{k|i} - \log \gamma\right|$ using bisection method;

        Recalculate $p_{j|i}$ with new values of $\kappa_i$;

        Calculate $p_{ij} = \frac{p_{j|i} + p_{i|j}}{2N}$ and $q_{ij} = \frac{q_{j|i} + q_{i|j}}{2N}$;

        Calculate KL divergence $\mathcal{L} = \sum_{ij} p_{ij} \log \frac{p_{ij}}{q_{ij}}$;

        Minimise loss $\mathcal{L}$ by updating $\theta$;

    **end for**

    $l = l + 1$;

**end while**

---

---

**Algorithm 2** SKMPS exemplar selection

---

**Input:** Array of input images $\mathbf{X}$, $f$ backbone model, labelling budget $k$

Calculate embeddings of all images in the latent space $\mathbf{H} = f(\mathbf{X})$, and normalize to obtain $\hat{\mathbf{H}}$;

Cluster $\hat{\mathbf{H}}$ into $k$ clusters using $k$-means to find the cluster centroids $\mathbf{C}$;

Initialise empty list $L = \{\}$;

$i = 0$;

**while** $i < k$ **do**

    $j = \text{ArgMin}(\text{cosine distance}(\hat{\mathbf{H}}, \mathbf{C}_{i:}))$;

    $L = \{L, j\}$;

    $i = i + 1$;

**end while**

**Return:** List $L$ of the indices of the images to be labelled

---

---

**Algorithm 3** MoLAR training algorithm.

---

**Input:** Training dataset $D$, exemplar dataset $D_e$, $N_E$ number of epochs, $f$ backbone model, Aug(.) augmentation strategy, exemplars per class $b_e$ to use during training, label smoothing parameter $\alpha$, $\tau$ temperature parameter

Initialise MLP head $g_\theta$ using vMF-SNE intialisation.

$l = 0$;

**while** $l < N_E$ **do**

    Randomly split $D$ into $B$ mini-batches;

    **for** $(\mathbf{x}_b, \mathbf{y}_b) \in \{D_1, ..., D_b, ..., D_B\}$ **do**

        Select $b_e$ exemplars per class from $D_e$, to obtain $(\mathbf{x}_e, \mathbf{y}_e)$;

        Augment $\mathbf{X} = \text{Aug}(\mathbf{x}_b)$;

        Augment $\mathbf{X}_e = \text{Aug}(\mathbf{x}_e)$;

        Calculate $\mathbf{Z} = g_\theta(f(\mathbf{X}))$, normalize to obtain $\hat{\mathbf{Z}}$;

        Calculate $\mathbf{Z}_e = g_\theta(f(\mathbf{X}_e))$, normalize to obtain $\hat{\mathbf{Z}}_e$;

        Calculate $\phi = \text{OneHot}(\mathbf{y}_b)(1 - \alpha) + \alpha$;

        Calculate $\mathbf{P} = \text{Softmax}(\hat{\mathbf{Z}} \cdot \hat{\mathbf{Z}}_e^\top)\,\phi$;

        **if** labels $\mathbf{y}_b$ are known **then**

            Calculate loss $\mathcal{L} = \frac{1}{|D_b|} \sum_i^{|D_b|} H(\mathbf{y}_{bi}, \mathbf{P}_{i,:})$;

        **else**

            Split $\mathbf{P}$ to obtain the outputs for different views of the same image $\mathbf{P}^*, \mathbf{P}^+$

            Calculate pseudolabels $\mathbf{y}^* = \rho(\frac{\mathbf{P}^* + \mathbf{P}^+}{2})$;

            Calculate average prediction $\bar{p} = \sum_i^{|D|} \rho\left(\frac{\mathbf{P}^*_{i,:} + \mathbf{P}^+_{i,:}}{2}\right)$;

            Calculate loss $\mathcal{L} = \frac{1}{2|D_b|} \sum_i^{|D_b|} \left(H(\mathbf{y}^*_{i,:}, \mathbf{P}^+_{i,:}) + H(\mathbf{y}^*_{i,:}, \mathbf{P}^*_{i,:})\right) - H(\bar{p})$;

        **end if**

        Minimise loss $\mathcal{L}$ by updating $\theta$;

    **end for**

    $l = l + 1$;

**end while**

---

# B   Additional results

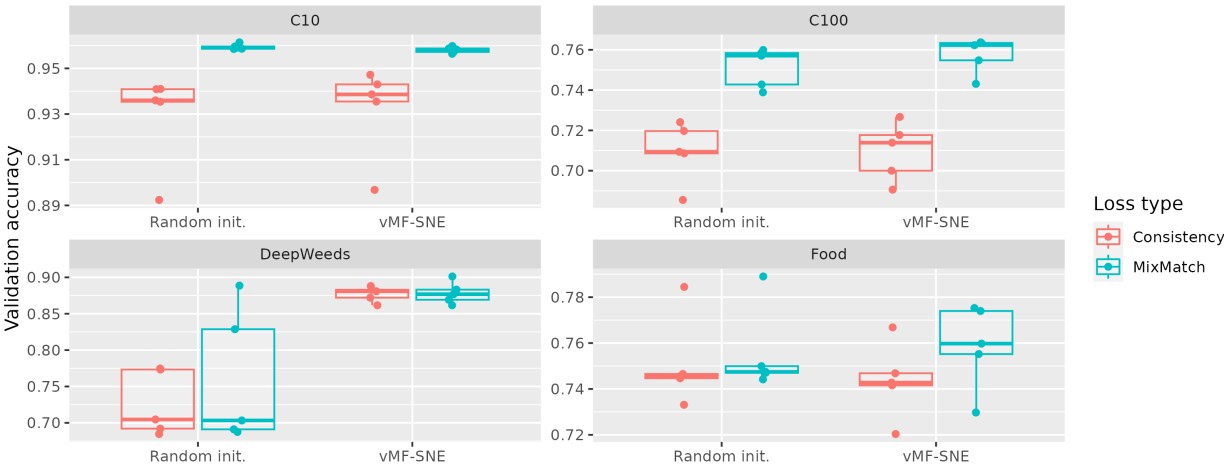

Figure S7: **Semi-supervised learning ablation study.** Comparison of the vMF-SNE initialisation and the averaging and sharpening (MixMatch) loss components of MoLAR-SS versus the original PAWS (Consistency) approach.

**Semi-supervised learning ablation studies.**   Fig. S7 shows that across the CIFAR-10, CIFAR-100, Food-101 and DeepWeeds datasets, using MoLAR-SS with vMF-SNE initialisation leads to the best overall results. For CIFAR-10 and CIFAR-100, the averaging and sharpening (MixMatch) loss significantly improves overall accuracy and the gain from vMF-SNE initialisation is less obvious. However, for DeepWeeds skipping vMF-SNE initialisation leads to poor results. In the case of the Food-101 dataset, the combination of both vMF-SNE initialisation and the MixMatch loss leads to the best average accuracy across all training runs.

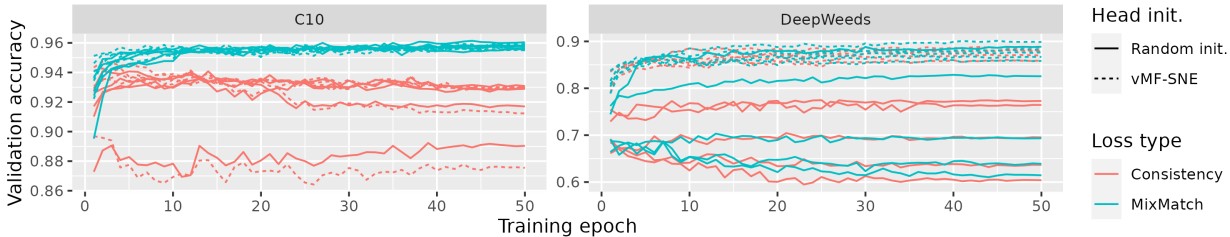

Figure S8: **Training progress curves.** Training curves comparing the vMF-SNE and random initialisation of the projection head and consistency versus the MoLAR-SS (Mixmatch) loss.

**Consistency based loss can degrade the performance of PAWS during a training run.**   In Fig. S8 the training curves for two of the datasets from Fig. S7 are shown. These training curves demonstrate that the consistency loss as used in PAWS can result in performance degrading for CIFAR-10 after around 10 epochs. When using the averaging and sharpening (MixMatch) loss as done in MoLAR-SS, the validation accuracy remains stable once the models have converged. This is likely caused when different views on the class boundaries have highly confident, but different, sharpened class probabilities. For PAWS this would result in different views of the same image being given different labels during training, resulting in instability at the class boundary. However, in MoLAR-SS this would result in one class or the other being chosen, or a more uncertain result, providing a more consistent supervision signal.

It is also noted that the performance improvement from vMF-SNE initialisation is not due to a head-start in training. While it does provide a better starting point for learning, in both cases models are trained until

they completely converge, and vMF-SNE initialised projection heads converge to higher performing maxima, particularly for the DeepWeeds dataset where training using a randomly initialised projection head leads to poor performance.

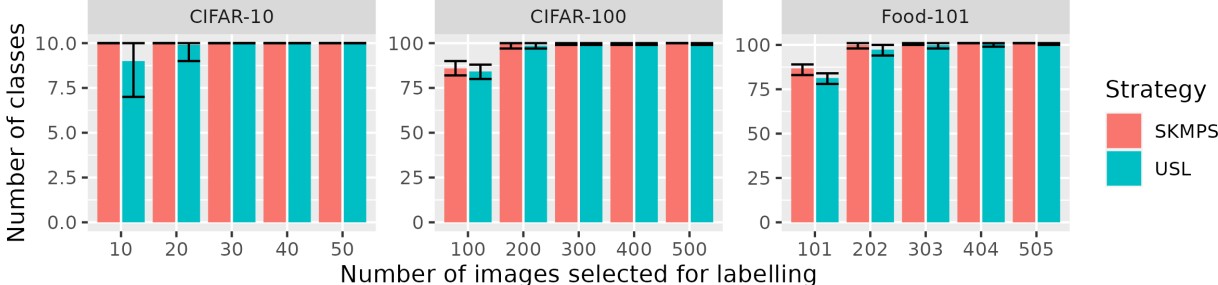

(a) **Number of classes.** Average number of classes sampled over 20 runs. Error bars show maximum and minimum number of classes sampled.

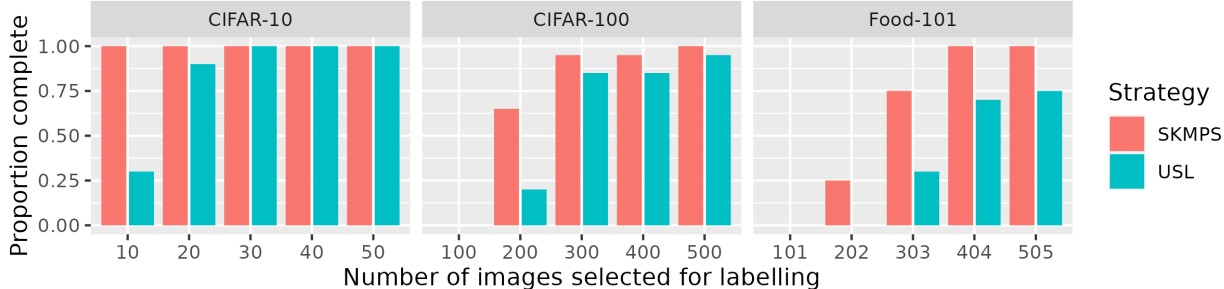

(b) **Proportion of classes.** Proportion of sets sampling all classes over 20 runs.

Figure S9: **Exemplar selection diversity.** Class diversity captured by different exemplar selection strategies.

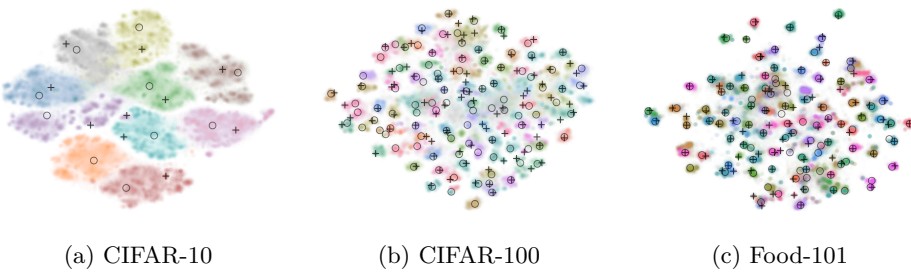

(a) CIFAR-10        (b) CIFAR-100        (c) Food-101

Figure S10: **Visualizing improvements in exemplar selection.** $t$-SNE visualisation of the DINOv2 ViT-S/14 backbone embeddings overlayed with the exemplars selected by the USL (+) and SKMPS (◦) strategies.

**SKMPS selects a more diverse set of images in comparison to USL.** In Fig. S9 we compare the diversity of classes sampled for the USL and SKMPS strategies with progressively larger budgets for the CIFAR-10, CIFAR-100 and Food-101 datasets. In all three cases, the SKMPS strategy samples a greater diversity of images, as measured by the number of classes sampled from each dataset for a given budget. Fig. S10 shows several examples of SKMPS sampling a more diverse set of exemplars in comparison to USL, when these are visualised using a $t$-SNE transformation.

When some classes are not sampled, this reduces the maximum attainable performance for a model as classes that are not represented by atleast one labelled image cannot be learned in most semi-supervised

learning frameworks. The USL approach considers density to be an indicator of exemplar informativeness, and preferentially chooses exemplars from denser regions of the embedding space of the foundation model (Wang et al., 2022b). In contrast, SKMPS only considers the density of the points through the $k$-means clustering process, which results in a more diverse sample while also being more computationally efficient.

Table S8: **Semi-supervised OOD detection with exemplars: Near OOD.** Comparison of semi-supervised representational approaches using a DINOv2 ViT-S/14 frozen backbone on OpenOOD benchmark datasets (Yang et al., 2022) with a KNN metric (Sun et al., 2022), employing a consistent set of exemplars obtained with SKMPS. We select 40 exemplars for CIFAR-10 and 400 for CIFAR-100. The ↑ means larger values are better and the ↓ means smaller values are better.

| IDD | Method | Near OOD Datasets | | | | | | Average | |
|---|---|---|---|---|---|---|---|---|---|
| | | CIFAR-10 | | CIFAR-100 | | Tiny ImageNet | | | |
| | | AUROC↑ | FPR↓ | AUROC↑ | FPR↓ | AUROC↑ | FPR↓ | AUROC↑ | FPR↓ |
| CIFAR-100 | PAWS | 90.11 | 51.13 | | | 90.99 | 46.54 | 90.55 | 48.84 |
| | MoLAR-SS | **95.79** | **26.05** | | | **94.60** | 32.30 | **95.19** | **29.18** |
| | *backbone* | 85.51 | 60.16 | | | 91.13 | **30.6** | 88.32 | 45.38 |
| CIFAR-10 | PAWS | | | 82.10 | 73.90 | 83.19 | 69.49 | 82.64 | 71.70 |
| | MoLAR-SS | | | 91.61 | 43.36 | 91.12 | 42.78 | 91.37 | 43.07 |
| | +vMF-SNE | | | 92.84 | 40.43 | 93.50 | 34.92 | 93.17 | 37.68 |
| | +SKMPS | | | **93.34** | **35.94** | 92.83 | 33.30 | 93.09 | 34.62 |
| | *backbone* | | | 90.66 | 40.13 | **97.16** | **11.12** | **93.91** | **25.62** |

Table S9: **Semi-supervised OOD detection with exemplars: Far OOD.** Comparison of semi-supervised representational approaches using a DINOv2 ViT-S/14 frozen backbone on OpenOOD benchmark datasets (Yang et al., 2022) with a KNN metric (Sun et al., 2022), employing a consistent set of exemplars obtained with SKMPS. We select 40 exemplars for CIFAR-10 and 400 for CIFAR-100. The ↑ means larger values are better and the ↓ means smaller values are better.

| IDD | Method | Far OOD Datasets | | | | | | | | Average | |
|---|---|---|---|---|---|---|---|---|---|---|---|
| | | DTD | | MNIST | | SVHN | | Places365 | | | |
| | | AUROC↑ | FPR↓ | AUROC↑ | FPR↓ | AUROC↑ | FPR↓ | AUROC↑ | FPR↓ | AUROC↑ | FPR↓ |
| CIFAR-100 | PAWS | 93.59 | 35.71 | 97.30 | 13.95 | 94.33 | **32.66** | 92.78 | 39.65 | 94.50 | 30.49 |
| | MoLAR-SS | 96.55 | 22.29 | **98.96** | **2.27** | **94.35** | 33.39 | 95.22 | 28.58 | **96.27** | **21.63** |
| | *backbone* | **98.92** | **4.8** | 87.98 | 86.15 | 72.76 | 85.57 | **97.03** | **12.18** | 89.17 | 47.18 |
| CIFAR-10 | PAWS | 91.36 | 48.71 | 96.44 | 15.37 | 84.99 | 81.74 | 89.27 | 54.24 | 90.52 | 50.01 |
| | MoLAR-SS | 95.44 | 27.84 | 98.67 | 5.35 | 93.81 | 39.49 | 94.48 | 27.4 | 95.6 | 25.02 |
| | +vMF-SNE | 96.05 | 24.95 | 98.68 | 1.91 | 95.50 | 28.77 | 96.06 | 22.20 | 96.57 | 19.46 |
| | +SKMPS | 97.45 | 15.62 | **99.78** | **0.00** | **98.04** | **10.78** | 96.70 | 17.13 | **97.99** | **10.88** |
| | *backbone* | **99.96** | **0.16** | 99.4 | 1.91 | 80.57 | 71.18 | **98.8** | **4.17** | 94.68 | 19.36 |

**Further OOD detection results.** In Table S8 and Table S9 we present the full set of results for each dataset in the OpenOOD benchmark, for the semi-supervised learning approaches that were considered. In general we find that the backbone embedding performs better on the datasets that are included in the DINOv2 training set, such as ImageNet, DTD and Places365 (which is included in the ADE20K dataset (Zhou et al., 2017b)). However, the backbone performs very poorly on OOD detection for datasets it hasn't seen before such as CIFAR-10, CIFAR-100, MNIST and SVHN, while the MoLAR-SS projection head can still perform well. Overall, the backbone performs slightly better on the near OOD CIFAR-10 benchmark (+0.82), while the MoLAR-SS embeddings perform better on the far OOD CIFAR-10 (+3.31) and both CIFAR-100 (+6.87/+7.10) benchmarks. Similar trends are observed for the supervised cases in Table S10, Table S11, Table S12, Table S13 and Table S14.

**Projection heads initialised using the vMF-SNE approach perform better than random initialisation, even if they are initialised using a different dataset.** In (Table S15) we fit MoLAR-SS models on the DeepWeeds dataset randomly initialising the projection head, and then use projection heads

Table S10: **Supervised OOD detection: Near OOD.** Comparison of representational approaches using a DINOv2 ViT-S/14 frozen backbone on OpenOOD benchmark datasets (Yang et al., 2022) with a KNN metric (Sun et al., 2022). The ↑ means larger values are better and the ↓ means smaller values are better.

| IDD | Method | Near OOD Datasets | | | | | | Average | |
| --- | --- | --- | --- | --- | --- | --- | --- | --- | --- |
| | | CIFAR-10 | | CIFAR-100 | | Tiny ImageNet | | | |
| | | AUROC↑ | FPR↓ | AUROC↑ | FPR↓ | AUROC↑ | FPR↓ | AUROC↑ | FPR↓ |
| CIFAR-100 | CIDER | 76.7 | 79.32 | | | 76.35 | 76.72 | 76.52 | 78.02 |
| | PALM | 87.32 | 55.58 | | | 85.44 | 55.49 | 86.38 | 55.54 |
| | MoLAR | **95.08** | **30.15** | | | **94.7** | **28.95** | **94.89** | **29.55** |
| | *backbone* | 87.97 | 56.05 | | | 91.82 | 29.63 | 89.9 | 42.84 |
| CIFAR-10 | CIDER | | | 93.88 | 33.38 | 90.96 | 32.46 | 92.42 | 32.92 |
| | PALM | | | 95.13 | 24.46 | 93.47 | 25.02 | 94.3 | 24.74 |
| | MoLAR | | | **95.17** | **23.29** | 95.72 | 16.98 | 95.44 | 20.14 |
| | *backbone* | | | 94.08 | 29.27 | **97.58** | **10.3** | **95.83** | **19.78** |

Table S11: **Supervised OOD detection: Far OOD.** Comparison of representational approaches using a DINOv2 ViT-S/14 frozen backbone on OpenOOD benchmark datasets (Yang et al., 2022) with a KNN metric (Sun et al., 2022). The ↑ means larger values are better and the ↓ means smaller values are better.

| IDD | Method | Far OOD Datasets | | | | | | | | Average | |
| --- | --- | --- | --- | --- | --- | --- | --- | --- | --- | --- | --- |
| | | DTD | | MNIST | | SVHN | | Places365 | | | |
| | | AUROC↑ | FPR↓ | AUROC↑ | FPR↓ | AUROC↑ | FPR↓ | AUROC↑ | FPR↓ | AUROC↑ | FPR↓ |
| CIFAR-100 | CIDER | 89.05 | 56.77 | **99.92** | **0.0** | **97.4** | **15.23** | 78.28 | 77.36 | 91.16 | 37.34 |
| | PALM | 94.74 | 29.93 | 99.8 | **0.0** | 97.19 | 16.12 | 90.88 | 43.46 | 95.65 | 22.38 |
| | MoLAR | 97.59 | 14.34 | 99.3 | 2.1 | 96.81 | 17.2 | 95.83 | 23.04 | **97.38** | **14.17** |
| | *backbone* | **97.89** | **8.48** | 92.25 | 52.51 | 82.32 | 82.49 | **96.19** | **15.95** | 92.16 | 39.86 |
| CIFAR-10 | CIDER | 97.25 | 14.4 | 99.58 | 0.04 | 99.3 | 1.07 | 94.07 | 22.99 | 97.55 | 9.62 |
| | PALM | 98.34 | 7.57 | 99.98 | **0.0** | **99.73** | **0.45** | 96.5 | 14.04 | 98.64 | 5.52 |
| | MoLAR | 98.71 | 5.39 | **99.99** | **0.0** | 99.47 | 1.14 | 98.34 | 6.77 | **99.13** | **3.33** |
| | *backbone* | **99.96** | **0.18** | 99.88 | 0.01 | 93.65 | 39.63 | **98.88** | **4.45** | 98.09 | 11.07 |

Table S12: **Supervised OOD detection with exemplars: Near OOD.** Comparison of representational approaches using a DINOv2 ViT-S/14 frozen backbone on OpenOOD benchmark datasets (Yang et al., 2022) with a KNN metric (Sun et al., 2022), employing a consistent set of exemplars obtained with SKMPS. We select 40 exemplars for CIFAR-10 and 400 for CIFAR-100. The ↑ means larger values are better and the ↓ means smaller values are better.

| IDD | Method | Near OOD Datasets | | | | | | Average | |
| --- | --- | --- | --- | --- | --- | --- | --- | --- | --- |
| | | CIFAR-10 | | CIFAR-100 | | Tiny ImageNet | | | |
| | | AUROC↑ | FPR↓ | AUROC↑ | FPR↓ | AUROC↑ | FPR↓ | AUROC↑ | FPR↓ |
| CIFAR-100 | CIDER | 79.8 | 67.93 | | | 77.32 | 71.92 | 78.56 | 69.92 |
| | PALM | 86.58 | 57.54 | | | 83.73 | 59.41 | 85.16 | 58.48 |
| | MoLAR | **95.58** | **29.37** | | | **95.34** | **29.4** | **95.46** | **29.39** |
| | *backbone* | 85.51 | 60.16 | | | 91.13 | 30.6 | 88.32 | 45.38 |
| CIFAR-10 | CIDER | | | 92.51 | 44.27 | 88.87 | 46.74 | 90.69 | 45.5 |
| | PALM | | | 92.79 | 42.25 | 90.53 | 44.52 | 91.66 | 43.38 |
| | MoLAR | | | 89.82 | 47.86 | 88.78 | 48.0 | 89.3 | 47.93 |
| | +vMF-SNE | | | 89.26 | 51.06 | 89.93 | 45.45 | 89.59 | 48.26 |
| | +SKMPS | | | **93.83** | **33.79** | 94.87 | 26.06 | **94.35** | 29.93 |
| | *backbone* | | | 90.66 | 40.13 | **97.16** | **11.12** | 93.91 | **25.62** |

initialised on various other datasets using the vMF-SNE approach. We found that initialising the projection head with vMF-SNE is better than random initialisation in every case, but best performance was obtained when initialisation was undertaken on DeepWeeds. Potentially, the vMF-SNE approach could be used as a

Table S13: **Supervised OOD detection with exemplars: Far OOD.** Comparison of representational approaches using a DINOv2 ViT-S/14 frozen backbone on OpenOOD benchmark datasets (Yang et al., 2022) with a KNN metric (Sun et al., 2022), employing a consistent set of exemplars obtained with SKMPS. We select 40 exemplars for CIFAR-10 and 400 for CIFAR-100. The ↑ means larger values are better and the ↓ means smaller values are better.

| IDD | Method | Far OOD Datasets | | | | | | | | Average | |
| --- | --- | --- | --- | --- | --- | --- | --- | --- | --- | --- | --- |
| | | DTD | | MNIST | | SVHN | | Places365 | | | |
| | | AUROC↑ | FPR↓ | AUROC↑ | FPR↓ | AUROC↑ | FPR↓ | AUROC↑ | FPR↓ | AUROC↑ | FPR↓ |
| CIFAR-100 | CIDER | 87.13 | 60.92 | **99.85** | **0.01** | **95.91** | **22.88** | 81.65 | 64.57 | 91.14 | 37.1 |
| | PALM | 93.08 | 39.43 | 99.71 | 0.09 | 95.82 | 23.47 | 89.41 | 48.27 | 94.51 | 27.82 |
| | MoLAR | 97.14 | 18.94 | 99.37 | 0.62 | 95.72 | 26.36 | 96.22 | 23.64 | **97.11** | **17.39** |
| | *backbone* | **98.92** | **4.8** | 87.98 | 86.15 | 72.76 | 85.57 | **97.03** | **12.18** | 89.17 | 47.18 |
| CIFAR-10 | CIDER | 95.77 | 27.23 | **99.98** | **0.0** | 99.25 | **1.23** | 92.15 | **36.04** | 96.79 | 16.13 |
| | PALM | 95.84 | 28.37 | 99.78 | **0.0** | **99.29** | 2.56 | 93.55 | 34.03 | 97.11 | 16.24 |
| | MoLAR | 94.21 | 34.29 | 96.83 | 15.45 | 92.32 | 44.9 | 91.97 | 39.08 | 93.83 | 33.43 |
| | +vMF-SNE | 93.94 | 35.55 | 99.14 | 0.88 | 93.17 | 39.44 | 93.58 | 32.69 | 94.96 | 27.14 |
| | +SKMPS | 96.98 | 17.3 | 99.79 | 0.0 | 98.04 | 9.52 | 97.04 | 15.18 | **97.96** | **10.5** |
| | *backbone* | **99.96** | **0.16** | 99.4 | 1.91 | 80.57 | 71.18 | **98.8** | **4.17** | 94.68 | 19.36 |

Table S14: **OOD detection with and without exemplars: ImageNet-200.** Comparison of representational approaches using a DINOv2 ViT-S/14 frozen backbone on ImageNet-200 OpenOOD benchmark datasets (Yang et al., 2022) with a KNN metric (Sun et al., 2022). We compare using the full training dataset from OOD detection, versus a consistent set of 600 exemplars obtained with SKMPS. The ↑ means larger values are better and the ↓ means smaller values are better.

| KNN Sample | Method | Near OOD Datasets | | | | Far OOD Datasets | | | | | |
| --- | --- | --- | --- | --- | --- | --- | --- | --- | --- | --- | --- |
| | | SSB-hard | | NINCO | | iNaturalist | | OpenImage-O | | DTD | |
| | | AUROC↑ | FPR↓ | AUROC↑ | FPR↓ | AUROC↑ | FPR↓ | AUROC↑ | FPR↓ | AUROC↑ | FPR↓ |
| Full dataset | CIDER | 83.22 | 56.74 | 90.13 | 44.72 | 97.15 | 17.52 | 95.79 | 24.36 | 95.61 | 17.15 |
| | PALM | 86.04 | 52.15 | **92.08** | 39.17 | **97.88** | **10.66** | 96.42 | 20.65 | 95.81 | 16.23 |
| | MoLAR | **89.68** | **42.77** | 91.68 | **38.32** | 97.59 | 13.94 | **96.54** | **20.09** | 96.04 | **15.22** |
| | *backbone* | 83.72 | 58.64 | 88.79 | 48.66 | 95.09 | 30.09 | 92.53 | 39.59 | 91.71 | 31.53 |
| 600 Exemplars | CIDER | 82.49 | 59.66 | 89.7 | 46.35 | 96.98 | 17.3 | 94.95 | 28.67 | 94.52 | 23.11 |
| | PALM | 85.00 | 55.63 | 91.47 | 41.62 | 97.46 | 13.96 | 95.62 | 25.97 | 94.90 | 20.64 |
| | MoLAR | 90.09 | 44.01 | 92.61 | 37.37 | **97.74** | **11.69** | **96.38** | **20.19** | 95.88 | 18.3 |
| | *backbone* | 84.67 | 52.27 | 89.49 | 44.48 | 97.98 | 10.66 | 95.98 | 22.2 | 95.10 | **16.97** |
| | PAWS | 87.46 | 51.62 | 90.29 | 48.77 | 96.09 | 23.24 | 94.39 | 32.18 | 94.71 | 23.36 |
| | MoLAR-SS | **91.33** | **42.74** | **93.59** | **36.57** | 97.25 | 16.2 | 95.67 | 25.47 | 95.86 | 21.42 |

Table S15: **Transfer learning with vMF-SNE.** Transferability of vMF-SNE initialisation between datasets.

| Method | Components | Dataset |
| --- | --- | --- |
| | Head init. | DeepWeeds |
| MoLAR-SS | Random init. | 76.0±9.3 |
| | vMF-SNE init. on C100 | 79.9±8.1 |
| | vMF-SNE init. on C10 | 81.5±7.1 |
| | vMF-SNE init. on Food | 84.2±6.0 |
| | vMF-SNE init. on DeepWeeds | **87.8**±1.5 |

method to initialise a projection head for a foundational model on a large dataset, and this could then be used to obtain better performance than random initialisation on smaller datasets.

**Naively incoporating elements of the PALM loss into PAWS and MoLAR-SS harms SSL performance.** It is shown in Table S16 that incorporating the loss term from PALM that encourages exemplars

Table S16: **Adding terms to increase class separation.** Adding terms from the PALM (Lu et al., 2024) loss that improve OOD detection performance to PAWS (Assran et al., 2021) and MoLAR-SS.

| Method | Components | | Dataset |
|---|---|---|---|
| | PALM (Lu et al., 2024) $\mathcal{L}_{\text{proto-contra}}$ | | C10 |
| PAWS | Y | | 92.1±0.1 |
| PAWS | N | | **92.9**±0.2 |
| MoLAR-SS | Y | | 94.2±0.1 |
| MoLAR-SS | N | | **95.9**±0.1 |

within a class to be compact, and far from exemplars of other classes, given by

$$
\mathcal{L}_{\text{proto-contra}} = -\frac{1}{CK} \sum_{c=1}^{C} \sum_{k=1}^{K} \log \frac{\sum_{k'=1}^{K} \mathbb{K}(k' \neq k) \exp\left(\hat{z}_c^{k'} \cdot \hat{z}_c^{k\top}/\tau\right)}{\sum_{c'=1}^{C} \sum_{k''=1}^{K} \mathbb{K}(k'' \neq k, c' \neq c) \exp\left(\hat{z}_{c'}^{k''} \cdot \hat{z}_{c'}^{k\top}/\tau\right)} \tag{15}
$$

results in poorer performance in comparison to not adding this loss term. Here $\mathbb{K}(\cdot)$ is an indicator function that avoids contrasting between the same exemplar, and $\hat{z}_c^k$ refers to the $k$th exemplar of class $c$, where $K$ exemplars belong to each of the $C$ classes.

## C  Additional results for vMF-SNE initialisation

### C.1  Performance

In a classification context, the performance of the backbone model can be measured non-parametrically using a weighted k-Nearest Neighbor (kNN) Classifier (Wu et al., 2018). This provides for a quantitative approach to measure the performance of the projection head initialised using vMF-SNE, as reproducing the local structure of the backbone model should provide a similar kNN accuracy. We find that for five of the seven datasets considered, we can meet or beat the kNN accuracy of the backbone model through vMF-SNE initialisation (Table S17).

### C.2  Hyperparameter ablation study

We undertook an ablation study to determine the sensitivity of vMF-SNE initialisation performance to the hyper-parameters within the algorithm. It was found that the method was highly insensitive to most parameters for the CIFAR-10 dataset, with only a significantly lower perplexity $\gamma$ or an order of magnitude higher concentration parameter $\tau$ resulting in changes to the kNN validation accuracy outside a single standard deviation (Table S18). The optimal parameter set appeared to be different between CIFAR-100 and the Food-101 datasets, with better performance for CIFAR-100 with a larger perplexity and smaller concentration and batch size, but better performance for Food-101 with a smaller perplexity and larger

Table S17: **vMF-SNE intialisation performance.** KNN accuracy of the model backbone versus the vMF-SNE initialised projection head.

| Embedding | Dataset | | | | | | |
|---|---|---|---|---|---|---|---|
| | C10 | C100 | EuroSAT | DeepWeeds | Flowers | Food | Pets |
| DINOv2 ViT-S/14 backbone | 96.2 | **82.3** | 90.0 | 88.7 | **99.3** | 80.9 | 91.5 |
| vMF-SNE proj. head | 96.3±0.1 | 79.0±0.1 | **91.9±0.1** | **91.3±0.1** | 98.2±0.5 | 81.0±0.1 | **93.0±0.1** |

Table S18: **vMF-SNE initialisation ablation study.** The results for the parameters used in the other results in this paper are reported as the mean KNN accuracy and standard deviation of five runs, while the other cases report a single run.

| Perplexity ($\gamma$) | C10 | C100 | Food |
|---|---|---|---|
| 5 | 96.1 | 74.0 | 81.7 |
| 30 | 96.3±0.1 | 79.0±0.1 | 81.0±0.1 |
| 50 | 96.3 | 79.5 | 80.9 |
| 100 | 96.3 | 80.1 | 80.7 |

| vMF concentration ($\tau$) | C10 | C100 | Food |
|---|---|---|---|
| 0.01 | 96.4 | 79.4 | 78.8 |
| 0.10 | 96.3±0.1 | 79.0±0.1 | 81.0±0.1 |
| 1.00 | 91.7 | 54.9 | 68.5 |

| $Z_2$ dimension | C10 | C100 | Food |
|---|---|---|---|
| 128 | 96.3 | 79.1 | 81.0 |
| 256 | 96.3 | 79.0 | 81.0 |
| 384 | 96.4 | 79.1 | 81.0 |
| 512 | 96.3±0.1 | 79.0±0.1 | 81.0±0.1 |
| 1024 | 96.3 | 79.0 | 81.0 |

| Batch size | C10 | C100 | Food |
|---|---|---|---|
| 128 | 96.2 | 79.5 | 80.4 |
| 256 | 96.3 | 79.1 | 80.7 |
| 512 | 96.3±0.1 | 79.0±0.1 | 81.0±0.1 |
| 1024 | 96.3 | 79.2 | 81.3 |

| Kernel | C10 | C100 | Food |
|---|---|---|---|
| vMF | 96.3±0.1 | | |
| Gaussian | 91.4±0.6 | | |

batch size. We also explored using a Gaussian kernel with CIFAR-10, and found that this resulted in much poorer performance.

# D    Further implementation details

## D.1    MoLAR, PAWS, CIDER and PALM

We adapted the PAWS and RoPAWS PyTorch (Paszke et al., 2019) implementation available at this GitHub repository to our own codebase using PyTorch-Lightning (Falcon & The PyTorch Lightning team, 2019) and Hydra (Yadan, 2019), to enable fair comparisons between PAWS and MoLAR-SS. We further adapted the official CIDER and PALM GitHub implementations to our codebase, and used them as the basis for the CIDER and PALM results presented here with a DINOv2 backbone to compare to MoLAR.

All of the MoLAR, MoLAR-SS and PAWS models that are fitted in this paper were trained for 50 epochs, and the vMF-SNE initialisation runs were trained for 20 epochs. This meant that even for the Food-101 dataset with 75,000 images, training a MoLAR or MoLAR-SS model took less than two hours with two Nvidia V100 32GB graphics cards for the ViT-S/14 and ViT-B/14 backbone, and around six hours for the ViT-L/14 backbone. For fair comparisons to the CIDER and PALM approaches, we trained these for 70 epochs to account for the additional vMF-SNE initialisation steps in MoLAR.

## D.2    vMF-SNE initialisation

We re-used the same augmentations, optimizer, scheduler and other details as for MoLAR and MoLAR-SS. The key difference is that the vMF-SNE initialisation approach only requires the unlabelled dataloader, compared to MoLAR-SS which requires both a exemplar and unlabelled dataloader. This meant vMF-SNE initialisation was slightly faster than training. We note than when using image augmentations for this approach, we treat each augmented view independently.

## D.3    $k$-means clustering and USL

We used the GPU-optimised $k$-means method available within cuML (Raschka et al., 2020). We save the normalised global class token from the DINOv2 model applied to each image of the dataset using the validation transformations, and run the $k$-means clustering method over this matrix. This method is extremely fast and can easily process millions of rows. We use the best clustering result from ten runs, and choose the number of clusters according to the number of images we wish to label. To select the exemplar to label from each cluster, we choose the image with the largest cosine similarity to the cluster centroid.

To reproduce the USL (Wang et al., 2022b) approach, we used the code published by the authors on GitHub. There are quite a few hyperparameters that are used within the USL method, so we selected the set that was used for ImageNet with CLIP (Radford et al., 2021), another foundation model based on the ViT architecture, for all of our experiments.

