# OpenReview forum: "A Mixture of Exemplars Approach for Efficient Out-of-Distribution Detection with Foundation Models"
_TMLR — Accepted by TMLR_

### Review · Reviewer_4wHB · 2025-06-10

**Summary Of Contributions:**

This work presents a new method for the detection of out-of-distribution (OOD) images. Typical OOD detection methods, used in computer vision, are based upon the distance between in-distribution (ID) and OOD image embeddings. Recent methods, improve such approaches by utilising von Mises-Fisher (vMF) mixture models, which cluster embeddings from the same target variable and maximise distance between embeddings of different classes.

This work introduces MoLAR, which also utilises vMF models, but uses images representative of the training data (so called, exemplars) to define the cluster centres in the vMF model. This reduces training times and the computational complexity of the optimisation algorithm, and allows for vMF clusters - the exemplars - to be used to define the distance metric that determines whether an image is ID or OOD. The latter means that class prediction and OOD detection require the same computational resource.

This work demonstrates the performance of MoLAR using the OpenOOD benchmark evaluations, and is reported to outperform two recent methods, PALM and CIDER, in a supervised setting, and PAWS in a semi-supervised setting. An ablation study is provided which motivates the choice of two key components of MoLAR i) a stochastic neighbour embedding initialisation for the projection head used to determine OOD samples and ii) the chosen selection strategy for the exemplars representative of the dataset, and shows that both lead to model performance.

Overall, this work provides interesting insight into a new approach for OOD detection, which can be combined with high performing foundation models, and requires no extra resource than class prediction for OOD detection.

**Audience:**

Yes

**Broader Impact Concerns:**

I believe in this case a broader impact statement is not necessary.

**Claims And Evidence:**

Yes

**Requested Changes:**

Overall, this is interesting work and will likely be a useful reference for others in OOD detection. Revisions addressing clarity and presentation would improve this submission.

Critical proposed adjustments:
- Please add a discussion of the limitations of MoLAR to the conclusion

Suggested adjustments
- Remove Fig 4. - it seems that the message of this figure is that initialisation leads to improved OOD detection via a better clustered embedding space. I think this figure doesn’t provide strong evidence for this claim - however that claim is supported by Table 6.
- Consider moving Figure 5 and 6 into the results section.
- On page 10 “However, in MoLAR-SS this would result in one class or the other being chosen, or a more uncertain result, providing a more consistent supervision signal” - this is not clear, please can you clarify this statement.
- In the abstract, “In some cases, only using the exemplars actually improves performance with MoLAR” - please consider removing or revising this statement as from the abstract alone it is not clear what this means.

**Strengths And Weaknesses:**

The submission provides good detail and introduction to the issues with OOD detection, and motivates the need for an algorithm such as MoLAR. There are many experiments included, providing insight into many facets of the performance of MoLAR and SKMPS (the exemplar selection strategy). The use of OpenOOD makes the results of this work easy to compare with past and future results, and also improves the reproducibility of this work.

Where the paper is weakest is in its clarity and organisation. On clarity, the paper throughout uses a KNN OOD metric cited from Sun et al. 2022, but it is not detailed in the main text how exactly this metric is computed, which makes the main text less self-contained. On page 8, statements are made about GPU time - it is not clear how many V100s are used for MoLAR, nor A100s for SemiReward in the main text.

On organisation, while I would praise the introduction, related work, methodology, and experiments section, I feel that the discussion and conclusion section could be improved. The discussion section, in my view, reads as an additional results section, and doesn't contextualise the results of the experiment section within the existing literature. Furthermore, the conclusion lacks a discussion of the limitations of this work.

---

> ### Author Response · Authors · 2025-06-29
> **Response**
>
> Thank you for your thorough review of this work. We provide a response to your comments below, and resulting changes to the text are marked in blue in the revised manuscript.
>
> > **1: please add a discussion of the limitations of MoLAR to the conclusion.**
>
> We have added a limitations section to the paper.
>
> > **2: remove figure 4.**
>
> We have kept this figure, but additionally added a reference to the tables that support this conclusion as noted.
>
> > **3: consider moving Figure 5 and 6 into the results section.**
>
> We have made this change in the text.
>
> > **4: please clarify statement on page 10.**
>
> We have changed this to “MoLAR-SS uses the same potential label for each view of an image providing a more consistent supervision signal.” which will hopefully be clearer for the reader.
>
> > **5. removing statement from the abstract.**
>
> We have removed this statement from the abstract.
>
> > **Other changes.**
>
> We have clarified the number of GPUS in our statements on page 8. We have further detailed the KNN metric and how it is used in the paper. We have also added a discussion and moved these results to the results section.

---

> > ### Comment · Reviewer_4wHB · 2025-07-07
> > **Response to authors**
> >
> > Thank you, to the authors, for responding to each of my comments.
> >
> > Reading the other reviews, I see there is some overlap in our constructive suggestions to improve the paper. I am happy to see an explanation of the KNN metric in Sec 4.1. I think this section would be further improved if it was expressed as a mathematical formula.
> >
> > I am also happy to see a discussion of limitations now included in this work. I would also encourage the authors to ask whether there are any scenarios where the assumptions underpinning MOLAR are not appropriate - for example, are there datasets where it is difficult to create suitable exemplars, that would prohibit applying MOLAR to a dataset?
> >
> >
> > I would be curious to hear if reviewer 7C17 feels that if response 1 to their review addresses their concerns.

---

> > > ### Author Response · Authors · 2025-07-11
> > > **Response**
> > >
> > > Thank you for your very constructive feedback. We've added the formula for the KNN OOD metric to 4.1 to make the description more concrete. We have further added a paragraph to the limitations section on instances where selecting suitable exemplars might be challenging for MoLAR as presented in this paper, and suggested alternative strategies that could be employed. In general, we expect MoLAR can be successful on any dataset where a linear head also provides good results, as they both utilise the same frozen features from the foundation model. We have add this as a further note to the limitations section.

---

### Review · Reviewer_7C17 · 2025-06-19

**Summary Of Contributions:**

To address OOD detection challenges, this work introduces a computationally efficient approach that optimally leverages frozen pretrained foundation models as classifier backbones. MoLAR consistently outperforms comparable approaches in OOD detection across extensive experimental evaluations.

**Audience:**

Yes

**Broader Impact Concerns:**

There is no outstanding ethical concerns.

**Claims And Evidence:**

No

**Requested Changes:**

I suggest the authors to address the following issues.

1. Explain why and how this method is significantly different from existing prototype based methods.

2. Explain the OOD scoring method

3. Add additional explanation and analysis for the results.

**Strengths And Weaknesses:**

Strength:
1. The paper provides a clear description of the methodology, with adequately designed experiments and reasonably thorough analysis.

Weakness:
1. The proposed method is composed of very simple training objective. The idea of measuring the similarity of samples and prototypes (examplars) and training a classifier using the similarity as logit has been widely explored by existing literature [A,B]. It is not clear how does the proposed method differ fundamentally from these works.

[A] Sun, Zhuohao, et al. "Classifier-head informed feature masking and prototype-based logit smoothing for out-of-distribution detection." IEEE Transactions on Circuits and Systems for Video Technology 34.7 (2024): 5630-5640.
[B] Gong, Mingrong, et al. "Out-of-distribution detection with prototypical outlier proxy." Proceedings of the AAAI Conference on Artificial Intelligence. Vol. 39. No. 16. 2025.


2. The description of OOD scoring is missing. Although the training objective is explained in Eq(6) and Eq(7), it is unclear how the OOD score is eventually calculated for testing samples.

3.	While Table S17 presents the ablation study of different hyperparameters, the main text appears to contain no discussion regarding the rationale for hyperparameter selection. It is unclear whether the same hyperparameter configurations were fixed across all datasets and models.
4.	Both abstract and introduction have mentioned "30 times faster inference", but in experiments lack a table to clarify how much inference time is required for different methods.
5.	The results in Table 2 suggest that semi-supervised learning does not improve the efficacy of OOD detection. The results in Table 3 further imply that FreeMatch could be a very competitive method which achieves comparable performance with a very small backbone network.

---

> ### Author Response · Authors · 2025-06-29
> **Response**
>
> Thank you for your thorough review of this work. We provide a response to your comments below, and resulting changes to the text are marked in blue in the revised manuscript.
>
> > **1: it is not clear how the proposed method differs fundamentally from other works.**
>
> Thank you for bringing to our attention the similarity of the approaches considered in this paper with prototypical methods. We have added a section to the discussion highlighting how PAWS, CIDER, PALM and MoLAR can be considered prototypical approaches, and comparing their vMF approach to [A] and [B].
>
> While [A] and [B] report results using ResNet backbones, making them not directly comparable to this work, we can compare results in relation to PALM, which MoLAR is found to improve upon. We show the AUROC results for CIFAR-10 with ResNet-18 below for the different models. PALM performs very strongly in comparison to the cited [A] and [B] baselines.
>
> |	| SVHN |  Places365 | LSUN | iSUN | Textures
> | -------- | ------- | ------- | ------- | ------- | ------- |
> | [A]  |  96.94 | 89.17 | 98.70 |94.19 |92.79 |
> | [B] |  98.45 | **95.65** | | |96.50 |
> | PALM	| **99.91**|94.80|**99.65**|**95.17**|**98.29**|
>
> > **2: the description of OOD scoring is missing.**
>
> We have added a paragraph to section 4.1 describing the OOD score.
>
> > **3: no discussion regarding the rationale for hyperparameter selection.**
>
> We have added a paragraph to section 4.1 providing further details on the hyperparameters used in the paper.
>
> > **4: experiments lack a table to clarify how much inference time is required for different methods.**
>
> Thank you for raising this. We have added Table 3 to provide these details, and adjusted this claim to reflect these more detailed timings.
>
> > **5: semi-supervised learning does not improve the efficacy of OOD detection.**
>
> We do not believe we make this claim in the paper. In semi-supervised learning the class label of the unlabelled images needs to be estimated during training, making tight clustering of the data in feature space more challenging.

---

> > ### Author Response · Authors · 2025-06-30
> > **Point 5**
> >
> > We’ve taken a look at #5 again, and we realised we may have misunderstood your comment.
> >
> > In Table 2, we compare MoLAR-SS with other semi-supervised approaches (PAWS), and bold these results as they are better. We do not compare MoLAR-SS with MoLAR, and rather expect it would be worse as only a small amount of labelled data is used in MOLAR-SS. We have added a note to the caption of Table 2 to address the confusion of having two bolded rows.
> >
> > We have taken a second look at the FreeMatch results in Table 3 and added the reported results for CIFAR-100. It performs quite well on CIFAR-10 with a very small network, as CIFAR-10 is not a particularly hard dataset for semi-supervised learning. However, it performs very poorly on CIFAR-100, which is more challenging.
> >
> > We hope this better addresses your concerns.

---

### Review · Reviewer_JmLB · 2025-06-25

**Summary Of Contributions:**

The paper presents an initialization and training procedure to train image classifiers from frozen foundation models that induces the capability of the model to classify whether an unseen sample is In-Distribution (ID) or Out-of-Distribution (OOD). Whereas previous approaches require the entire training set for OOD detection, the proposed methodology allows for using only a select subset of “exemplars” reducing inference cost. They also adapt their approach to be able to work in semi-supervised learning settings. Additionally, they propose an exemplar selection and projection head initialization methodology that outperforms prior common approaches. Results outperform previous state-of-the-art and ablation experiments were done on the main components.

**Audience:**

Yes

**Broader Impact Concerns:**

None that comes to mind.

**Claims And Evidence:**

Yes

**Requested Changes:**

**Related to W1**

[RC1] A bit of context / explanation of why vMF components differ from related approaches is missing.

[RC2] The KNN metric used for evaluation in the table is not well defined, it would be necessary to have a self-contained explanation in this manuscript since it is the basis for lots of the experimentation. Specifically, which embedding space is used for each of the architecture? It measures a distance, but the table seems to report percentages? Is there thresholding applied to classify ID or OOD?

[RC3] Could you provide an Algorithm-form presentation of how Molar would do OOD-detection? (Like Algorithm 1 or 2)

**Related to W2**

[RC4] Improvements when using fewer examples (full dataset vs SKMPS picked) is surprising to me, could this be noise? Were Tables 1 & 2 run with multiple seeds? If not I would ask for that.

**Related to W3**

[RC5] In the “Examplar selection strategies” second paragraph, it is ambiguous whether SKMPS is running the (i), (ii), and (iii) steps for each label or the entire dataset. It becomes clearer in the results section, but it would be good to add the clarification there.

[RC6] “the projection head cannot be initialized as an identity mapping”, could you provide a sentence or a reference that clarifies why that is the case.

**Minor**

[RC7] Comparing Table 1 and Table 2 is a bit tedious, I think the presentation would greatly benefit from adding the differences with Table 1 somewhere in there. Could be in very small font.

[RC8] (Yang et al., 2023) reference is not included in the related works.

[RC9] Fix typo in Appendix B "SKMPS selects a more diverse set of images in comparsion to USL."

**Strengths And Weaknesses:**

**Strengths**

[S1] The paper is clearly written and well formatted.

[S2] Strong literature review. Love the interleaved on-the-fly related works format.

[S3] Well detailed experimentation

[S4] Results are convincing and ablation experiments were done

**Weaknesses**

[W1] The paper suffers a bit from assuming the reader is familiar with concepts from the related literature. I’ve made a couple suggestions in the requested changes to address this.

[W2] Some of the experimentation was done without error measurements.

[W3] Few gaps in the description of approach of SKMPS and the Projection head initialization.

---

> ### Author Response · Authors · 2025-06-29
> **Response**
>
> Thank you for your thorough review of this work. We provide a response to your comments below, and resulting changes to the text are marked in blue in the revised manuscript.
>
> > **RC1: further context on why vMF components differ from related approaches.**
>
> We have added further details on how vMF OOD detection methods differ to other prototypical OOD detection approaches in the discussion, and added further context around self-supervised representational OOD detection methods in the related works section.
>
> > **RC2 and RC3: further details on KNN OOD metric.**
>
> We have added a paragraph to section 4.1 describing the KNN OOD detection metric in full, providing all of these details. The algorithm for OOD detection is the same as described in Sun et al. 2022, so we omit including it.
>
> > **RC4: improvements when using fewer examples could be due to noise.**
>
> We have run OOD detection with multiple seeds for the one of the MoLAR cases that is likely to have the greatest noise (MoLAR-SS on CIFAR-10). Over five seeds, we find a mean 94.8±1.0 for near-OOD and 98.2±0.2 for far-OOD detection. This observed variance is similar to that found when training PALM and CIDER with different seeds in Lu et al. 2024. To run all of Table 1 and 2 with multiple seeds would be a significant undertaking, and we follow prior works in benchmarking with a single seed.
>
> Without cherry picking results (we note that the seed for CIFAR-10 was among the worst performing, with a 93.1 and 98.0 AUCROC for near and far OOD detection), we observe consistent and significant improvement over previous baselines. However, these variances could suggest that the improvements observed when using fewer examples may be due to noise, and we have amended the claims in the paper to reflect this.
>
> > **RC5: ambiguous whether SKMPS is running for each label or the entire dataset.**
>
> We have added further text clarifying it is for the entire dataset.
>
> > **RC6: clarify why the projection head cannot be initialized as an identity mapping.**
>
> We have added further text to explain this.
>
> > **RC7: comparing Table 1 and Table 2 is a bit tedious.**
>
> We have added an additional table to make this comparison clearer.
>
> [RC8] and [RC9] are fixed in the text.

---

> > ### Comment · Reviewer_JmLB · 2025-07-11
> > **Reviewers response**
> >
> > Thank you for addressing the request changes. All were adequately answered.

---

### Decision · Action_Editor_jt19 · 2025-07-28

**Recommendation:** Accept as is

**Additional Comments:**

This paper proposes a method for OOD detection. The reviewers found the paper to be well motivated, and all the concerns were addressed by the authors during the rebuttal. I thus recommend acceptance.

**Audience:**

Yes

**Audience Explanation:**

Yes, reviewers unanimously agree.

**Claims And Evidence:**

Yes

**Claims Explanation:**

Yes.

---

> ### Author Response · Authors · 2025-08-22
> **Thank you**
>
> I’d like to express our sincere thanks for your time and attention in managing our submission. We’re also grateful to the reviewers for their insightful comments and suggestions. Their feedback helped us sharpen our presentation and strengthen the quality of the work.
>
> Thank you again for your efforts and support. It means a lot to us, and we look forward to the opportunity to continue working with the community.